# Cervical cerclage for prevention of preterm birth and adverse perinatal outcome in twin pregnancies with short cervical length or cervical dilatation: A systematic review and meta-analysis

Francesco D'Antonio[1], Nashwa Eltaweel[2], Smriti Prasad[3], Maria Elena Flacco[4], Lamberto Manzoli[5], Asma Khalil[3,6,7,8]*

1 Center for Fetal Care and High-Risk Pregnancy, University of Chieti, Chieti, Italy, 2 Division of Biomedical Science, Warwick Medical School University of Warwick, University Hospital of Coventry and Warwickshire, Coventry, United Kingdom, 3 Fetal Medicine Unit, St George's Hospital, London, United Kingdom, 4 Department of Environmental and Preventive Sciences, University of Ferrara, Ferrara, Italy, 5 Department of Medical and Surgical Sciences, University of Bologna, Bologna, Italy, 6 Vascular Biology Research Centre, Molecular and Clinical Sciences Research Institute, St George's University of London, London, United Kingdom, 7 Twins Trust Centre for Research and Clinical Excellence, St George's Hospital, London, United Kingdom, 8 Fetal Medicine Unit, Liverpool Women's Hospital, University of Liverpool, Liverpool, United Kingdom

* akhalil@sgul.ac.uk

**Data Availability Statement:** All relevant data is available within the manuscript and Supporting Information files.

**Funding:** The author(s) received no specific funding for this work.

## Abstract

### Background

The optimal approach to prevent preterm birth (PTB) in twins has not been fully established yet. Recent evidence suggests that placement of cervical cerclage in twin pregnancies with short cervical length at ultrasound or cervical dilatation at physical examination might be associated with a reduced risk of PTB. However, such evidence is based mainly on small studies thus questioning the robustness of these findings. The aim of this systematic review was to determine the role of cervical cerclage in preventing PTB and adverse maternal or perinatal outcomes in twin pregnancies.

### Methods and findings

Key databases searched and date of last search: MEDLINE, Embase, and CINAHL were searched electronically on 20 April 2023.

Eligibility criteria: Inclusion criteria were observational studies assessing the risk of PTB among twin pregnancies undergoing cerclage versus no cerclage and randomized trials in which twin pregnancies were allocated to cerclage for the prevention of PTB or to a control group (e.g., placebo or treatment as usual). The primary outcome was PTB <34 weeks of gestation. The secondary outcomes were PTB <37, 32, 28, 24 weeks of gestation, gestational age at birth, the interval between diagnosis and birth, preterm prelabor rupture of the membranes (pPROM), chorioamnionitis, perinatal loss, and perinatal morbidity. Subgroup

**Competing interests:** The authors have declared that no competing interests exist.

**Abbreviations:** AR, absolute risk; CI, confidence interval; IPD, individual patient data; ISUOG, International Society of Ultrasound in Obstetrics and Gynecology; IVH, intraventricular hemorrhage; MC, monochorionic; MD, mean difference; MeSH, medical subject heading; NEC, necrotizing enterocolitis; NICU, neonatal intensive care unit; NOS, Newcastle–Ottawa scale; pPROM, preterm prelabor rupture of the membrane; PTB, preterm birth; RCT, randomized controlled trial; RDS, respiratory distress syndrome; ROP, retinopathy of prematurity; RR, risk ratio; sFGR, selective fetal growth restriction; TTTS, twin-to-twin transfusion syndrome.

analyses according to the indication for cerclage (short cervical length or cervical dilatation) were also performed.

Risk of bias assessment: The risk of bias of the included randomized controlled trials (RCTs) was assessed using the Revised Cochrane risk-of-bias tool for randomized trials, while that of the observational studies using the Newcastle–Ottawa scale (NOS).

Statistical analysis: Summary risk ratios (RRs) of the likelihood of detecting each categorical outcome in exposed versus unexposed women, and (b) summary mean differences (MDs) between exposed and unexposed women (for each continuous outcome), with their 95% confidence intervals (CIs) were computed using head-to-head meta-analyses.

Synthesis of the results: Eighteen studies (1,465 twin pregnancies) were included. Placement of cervical cerclage in women with a twin pregnancy with a short cervix at ultrasound or cervical dilatation at physical examination was associated with a reduced risk of PTB <34 weeks of gestation (RR: 0.73, 95% CI [0.59, 0.91], $p = 0.005$ corresponding to a 16% difference in the absolute risk, AR), <32 (RR: 0.69, 95% CI [0.57, 0.84], $p < 0.001$; AR: 16.92%), <28 (RR: 0.54, 95% [CI 0.43, 0.67], 0.001; AR: 18.29%), and <24 (RR: 0.48, 95% CI [0.23, 0.97], $p = 0.04$; AR: 15.57%) weeks of gestation and a prolonged gestational age at birth (MD: 2.32 weeks, 95% [CI 0.99, 3.66], $p < 0.001$). Cerclage in twin pregnancy with short cervical length or cervical dilatation was also associated with a reduced risk of perinatal loss (RR: 0.38, 95% CI [0.25, 0.60], $p < 0.001$; AR: 19.62%) and composite adverse outcome (RR: 0.69, 95% CI [0.53, 0.90], $p = 0.007$; AR: 11.75%). Cervical cerclage was associated with a reduced risk of PTB <34 weeks both in women with cervical length <15 mm (RR: 0.74, 95% CI [0.58, 0.95], $p = 0.02$; AR: 29.17%) and in those with cervical dilatation (RR: 0.68, 95% CI [0.57, 0.80], $p < 0.001$; AR: 35.02%). The association between cerclage and prevention of PTB and adverse perinatal outcomes was exclusively due to the inclusion of observational studies. The quality of retrieved evidence at GRADE assessment was low.

## Conclusions

Emergency cerclage for cervical dilation or short cervical length <15 mm may be potentially associated with a reduction in PTB and improved perinatal outcomes. However, these findings are mainly based upon observational studies and require confirmation in large and adequately powered RCTs.

---

### Author summary

#### Why was this study done?

- Twin pregnancies are at high risk of preterm birth (PTB).

- Recent evidence suggests that placement of cervical cerclage in twin pregnancies with short cervical length at ultrasound or cervical dilatation at physical examination might be associated with a reduced risk of PTB.

- However, such evidence is based mainly on small studies thus questioning the robustness of these findings.

## What did the researchers do and find?

- We performed a systematic review and meta-analysis to elucidate whether cervical cerclage in women with twin pregnancy with short cervical length or cervical dilatation may prevent PTB.

- We included 18 studies. The primary outcome was PTB <34 weeks of gestation.

- We found that cervical cerclage in women with short cervical length or cervical dilatation was associated with a reduced risk of PTB <34 weeks, gestational age at birth, and adverse neonatal outcome.

- The strength of association between cerclage and reduced risk of PTB was maintained when considering women with short cervix on ultrasound and those with cervical dilatation at physical examination separately.

## What do these findings mean?

- Cervical cerclage in twin pregnancies with short cervical length or cervical dilatation may be potentially associated with a reduced risk of PTB and improved neonatal outcomes.

- However, these findings are mainly based on observational studies and, to improve robustness of evidence, confirmation of these outcomes in large and appropriately designed randomized controlled trials (RCTs) is required.

## Introduction

Twin pregnancies are at increased risk of perinatal morbidity and mortality compared to singletons, primarily due to preterm birth (PTB), fetal anomalies, and complications unique to monochorionic (MC) placenta, such as twin-to-twin transfusion syndrome (TTTS) and selective fetal growth restriction (sFGR) [1–8]. The incidence of PTB in twin pregnancy has been reported to be approximately 20% in recent series and this risk differs according to the chorionicity and amnionicity. Around 60% of twin pregnancies deliver prior to 37 weeks and 12% before 34 weeks of gestation, with rates 5 and 8 times higher than the equivalent rates for a singleton pregnancy, respectively [9].

In singleton pregnancies with recognized risk factors for PTB, vaginal progesterone is the primary intervention with consistently demonstrated effectiveness in preventing PTB, followed by cervical cerclage [10]. However, observational studies and systematic reviews have reported a beneficial role of cervical cerclage in pregnancies with an extremely short cervix, defined as a cervical length of less than 10 mm on ultrasound scan [11].

Conversely, there is less evidence on the optimal strategy for preventing PTB in twin pregnancies. Several randomized trials and systematic reviews reported little or no benefit of vaginal progesterone, cerclage, or pessary in twin pregnancies [12–14]. However, these studies were limited by small sample size and large heterogeneity in their inclusion criteria, study populations, and outcomes observed. These limitations did not allow the authors to reach evidence-based conclusions on the role of these interventions in reducing the risk of PTB in twin

pregnancies. More importantly, in the last few years, an increasing number of studies reporting a potential beneficial role of cerclage in reducing the risk of PTB and adverse outcomes in twin pregnancies have been published [15–20]. These studies have challenged the prevailing view around the lack of effectiveness of cerclage in twin pregnancies.

We performed a systematic review and meta-analysis of the published literature to determine the role of cervical cerclage in preventing PTB and adverse maternal and perinatal outcomes in twin pregnancies.

## Methods

### Data sources

This review was performed according to an a priori designed protocol recommended for systematic reviews and meta-analysis [21–24]. MEDLINE, Embase, and CINAHL were searched electronically since inception on 6 July 2022 and updated on 20 April 2023 utilizing combinations of the relevant medical subject heading (MeSH) terms, keywords, and word variants for "twin pregnancies," "multiple pregnancies," "cerclage," and "preterm birth." The search and selection criteria with no language restriction. The search strategy is outlined in S1 Table. The reference lists of relevant articles and reviews were hand-searched for additional reports. The study was registered with the PROSPERO database (Registration number: CRD42022351058). This study is reported as per the Preferred Reporting Items for Systematic Reviews and Meta-Analyses (PRISMA) guidelines (S2 Table) [25].

### Eligibility criteria, main outcomes measures

Inclusion criteria were observational studies assessing the risk of PTB among twin pregnancies undergoing cerclage versus no cerclage and randomized trials in which twin pregnancies were allocated to cerclage for the prevention of PTB or to a control group (e.g., placebo or treatment as usual).

The primary outcome was PTB <34 weeks of gestation.

The secondary outcomes were:

- PTB <37 weeks

- PTB <32 weeks

- PTB <28 weeks

- PTB <24 weeks

- Gestational age at birth [weeks]

- Interval between diagnosis and birth [weeks]

- Preterm prelabor rupture of the membranes (pPROM), defined as the rupture of the membranes before labor and before 37 weeks of gestation

- Chorioamnionitis

- Perinatal loss, including miscarriage, intra-uterine, and neonatal death

- Apgar score <7 at 5 min

- Birthweight <2,500 grams

- Birthweight <1,500 grams

- Birthweight expressed as a continuous variable

- Respiratory distress syndrome (RDS)

- Intraventricular hemorrhage (IVH), grades III and IV

- Necrotizing enterocolitis (NEC)

- Retinopathy of prematurity (ROP)

- Neonatal sepsis

- Admission to the neonatal intensive care unit (NICU)

- Length of stay in NICU (days).

Both primary and secondary outcomes were explored first in women with either cervical dilatation at a physical examination or short cervix (<25 mm) at ultrasound and in those with short cervical length at ultrasound and cervical dilatation separately. Furthermore, we planned to perform subgroup analyses according to different cut-offs of cervical length at ultrasound (<25 mm, <15 mm, and <10 mm) and cervical dilatation at physical examination (<2 cm versus >2 cm), according to chorionicity and type of cerclage (McDonald versus Shirodkar). In the McDonald technique, a suture is placed around the cervix in purse-string fashion and securely tied anteriorly. Conversely, the Shirodkar technique requires a transverse incision in the vaginal mucosa of the anterior and posterior cervix to avoid injury of the bladder and rectum, respectively. The lateral angles of the anterior and posterior incisions are then expanded with blunt fingertip dissection of the lateral cervix and a woven thread is then passed through the submucosal tunnel from anterior to posterior on both sides of the cervix. After the suture is placed on both sides of the cervix, the knot is tied in the posterior defect.

## Data collection and analysis

Two reviewers (FDA, NA) independently extracted data. Inconsistencies were discussed among the reviewers and consensus reached. For those articles in which data on short cervical length was not reported separately for subgroups of women (<15 mm and 15 to 25 mm), but the methodology was such that the information might have been recorded initially, the authors were contacted, and the data requested.

The risk of bias of the included randomized controlled trial (RCTs) was assessed using the Revised Cochrane risk-of-bias tool for randomized trials (RoB 2) [26]. According to this tool, the risk of bias in each included study is judged according to 5 domains: bias arising from the randomization process, bias due to deviations from intended interventions, bias due to missing outcome data, bias in the measurement of the outcome, and bias in the selection of the reported result. Although the RoB2 tool does not provide an overall risk of bias assessment, the overall risk of bias was considered low if 4 or more domains were rated as low risk (not counting "other biases"), with at least one being sequence generation or allocation concealment, according to what was reported in previous systematic reviews of intervention.

The risk of bias in the observational studies was performed using the Newcastle–Ottawa scale (NOS) for cohort studies [27]. According to NOS, each study is judged on 3 broad perspectives: selection of the study groups, comparability of the groups, and ascertainment of the outcome of interest. Assessment of the selection of a study includes the evaluation of the representativeness of the exposed cohort, selection of the nonexposed cohort, ascertainment of exposure, and the demonstration that the outcome of interest was not present at the start of the study. Assessment of the comparability of the study includes the evaluation of the comparability of cohorts based on the design or analysis. Finally, ascertainment of the outcome of interest includes the evaluation of the type of assessment of the outcome of interest, and length

and adequacy of follow-up. According to NOS, a study can be awarded a maximum of 1 star for each numbered item within the selection and outcome categories. A maximum of 2 stars can be given for comparability [27]. The conclusions of the meta-analysis on the primary outcome were assessed using the GRADE approach by the first author, who was familiar with GRADE (GRADEpro, Version 20, 2014, McMaster University, Hamilton, Ontario, Canada) [28]. A second author verified the ratings; any disagreements were reconciled after discussion. The pooled analysis of the primary outcome was assessed in relation to the quality of the evidence scored in the 5 domains specified within GRADE: limitations in study design and/or execution (risk of bias), inconsistency of results, indirectness of evidence, imprecision of results, and publication bias [28].

## Statistical analysis

We examined a total of 17 maternal and perinatal outcomes, either categorical or continuous, in a sample of women with twin pregnancies at risk of PTB undergoing cerclage (exposed women) versus no cerclage (unexposed women). All analyses were performed 3 times: (a) including women undergoing cerclage for either cervical dilatation or short cervical length at ultrasound; (b) including only women with short cervical length at ultrasound; and (c) including only women undergoing cerclage for dilated cervix.

First, we performed head-to-head meta-analyses and computed (a) summary risk ratios (RR) of the likelihood of detecting each categorical outcome in exposed versus unexposed women; and (b) summary mean differences (MDs) between exposed and unexposed women (for each continuous outcome), with their 95% confidence intervals (CIs). The relative intra-study heterogeneity was quantified using the $I^2$ metric, and its 95% CIs were computed using the *heterogi* command in Stata. For categorical outcomes, data were combined using a random-effect generic inverse variance approach that enables the inclusion of diverse estimates of relative risk (i.e., OR and HR) into the same meta-analysis. From each paper, we extracted the adjusted estimates of each outcome, or, when these were not available, the unadjusted estimates. If a paper reported the results of different multivariate models, the most stringently controlled estimates (those from the model adjusting for more factors) were extracted. If different models controlled for the same number of covariates, the model containing the most relevant covariates was used for the analysis. In case different measures of risk were to be included in the same pooled analysis (e.g., OR and RR), the OR was converted into RR [29,30]. Furthermore, we stratified all analyses according to the study design (randomized controlled trial or observational).

Finally, in order to provide some estimates of the crude rates of each categorical outcome, we also performed meta-analysis of proportions, combining the data of women undergoing and not undergoing cerclage separately [29,30]. To account for between-study heterogeneity, the analyses were performed using a random-effect model.

Potential small study effect was assessed graphically, using funnel plots (displaying the ORs from individual comparisons versus their precision [1/SE]), and formally, using Egger's regression asymmetry test [31]. All analyses were carried out using RevMan 5.4 (The Cochrane Collaboration, 2020) [32] and Stata, version 13.1 (Stata Corp., College Station, TX, 2013).

## Results

### Study selection and characteristics

A total of 1,070 studies were identified, 60 were assessed with respect to their eligibility for inclusion, and 18 included in the systematic review (Table 1, Fig 1) [15,17,20,33–47]. A list of the excluded studies and reasons for their exclusion is provided in S3 Table. These 18 studies

**Table 1. General characteristics of the studies included in the systematic review.**

| Author | Year | Country | Study design | Period considered | Inclusion criteria | Type of cerclage | Gestational age at cerclage placement (weeks) | Adjusted analysis* | Primary outcome | Twin pregnancies (*n*) |
|---|---|---|---|---|---|---|---|---|---|---|
| Qiu [33] | 2023 | China | Observational | 2015–2021 | Twin pregnancies with cervical dilatation (1 cm) at 18–26 weeks | McDonald | 22, 8$^\Xi$ | Yes[a] | Gestational age at birth | 99 |
| Qiu [34] | 2022 | China | Observational | 2015–2021 | Twin pregnancies with short CL ≤25 mm at 18–26 weeks | McDonald | 22, 9 ± 1.7$^\S$ | Yes[a] | Gestational age at birth | 90 |
| Yao [20] | 2022 | China | Observational | 2014–2020 | Twin pregnancies with short CL ≤25 mm at 16–28 weeks | McDonald | 16–28$^\varsigma$ | Yes[b] | PTB <34 weeks | 320 |
| Zeng [35] | 2022 | China | Observational | 2015–2020 | Twin pregnancies with cervical dilatation and prolapsed membranes | McDonald | 16+0–26+6$^\varsigma$ | No | PTB <28 weeks | 97 |
| Pan [36] | 2020 | China | Observational | 2015–2019 | Twin pregnancies with asymptomatic cervical shortening or dilation at ultrasound and/or physical examination in mid-gestation | McDonald | 23.7 (14.14–25.86)$^\Psi$ | Yes[c] | Gestational age at birth | 62 |
| Wu [17] | 2020 | Taiwan | Observational | 2000–2017 | DCDA twin pregnancies with a short cervical length (25 mm] | McDonald | NR[1] | No | Gestational age at birth | 46 |
| Roman [15] | 2020 | US | RCT | 2015–2019 | asymptomatic cervical dilation from 1–5 cm between 16 0/7 to 23 6/7 weeks | McDonald | 20, 7 ± 1, 7$^\S$ | Yes[b] | PTB <34 weeks | 30 |
| Han [37] | 2020 | US | Observational | 2003–2016 | Twin pregnancies with history of prior preterm birth, ultrasound-identified short cervix ≤2.5 cm, and cervical dilation ≥1.0 cm at 14–26 weeks | Shirodkar | 20 (12–27)$^\Psi$ | Yes[d] | PTB <32 weeks | 135 |
| Qureshey [38] | 2022 | US | Observational | 2006–2016 | Twin pregnancies with short CL ≤25 mm at 15–24 | McDonald | 15–24$^\varsigma$ | Yes[b] | Gestational age at birth | 64 |
| Abbasi [33] | 2018 | Canada | Observational | 2003–2014 | Dilated cervix and intact membranes before 25–week gestation | McDonald | 21.5 ± 2.6$^\S$ | No | PTB <34 weeks | 36 |
| Adams [40] | 2018 | US | Observational | 2008–2014 | Twin gestations identified with cervical length of ≤2.5 cm before 24 weeks gestation | McDonald | 20.8 ± 1.9$^\S$ | Yes[e] | PTB <35 weeks | 82 |
| Houlihan [41] | 2016 | UK | Observational | 2006–2014 | DC twin pregnancies with an ultrasound-determined cervical length of 1–24 mm at 16–24 weeks | McDonald | NR[1] | Yes[f] | PTB <32 weeks | 80 |
| Roman [42] | 2016 | US | Observational | 1997–2014 | Twin pregnancies identified with cervical dilation of >1 cm at 16–24 weeks | McDonald | 20, 7 ± 1, 6$^\S$ | Yes[g] | PTB <34 weeks | 76 |
| Roman [43] | 2015 | US | Observational | 1995–2012 | Asymptomatic twin pregnancies with TVU CL 25 mm at 16–24 weeks | Shirodkar or McDonald | NR[1] | Yes[h] | PTB <34 weeks | 140 |

(*Continued*)

**Table 1.** (Continued)

| Author | Year | Country | Study design | Period considered | Inclusion criteria | Type of cerclage | Gestational age at cerclage placement (weeks) | Adjusted analysis* | Primary outcome | Twin pregnancies (*n*) |
|---|---|---|---|---|---|---|---|---|---|---|
| Roman [44] | 2005 | US | Observational | 1996–2002 | ALl twin pregnancies with CL ≤25 mm before 24 weeks | Shirodkar | 20.8 (15.7–23.6)[Ψ] | No | PTB <32 weeks | 31 |
| Newman [45] | 2002 | US | Observational | 1994–2001 | Twin pregnancies with short CL ≤25 mm at 18–26 | McDonald | 18–26[ϛ] | No | PTB | 33 |
| Althuisius [46] | 2001 | The Netherland-Australia | RCT | 1995–2000 | Twin pregnancies with short CL (≤25 mm) | McDonald | Before 27 weeks | No | PTB <34 weeks | 17 |
| Rust [47] | 2000 | US | RCT | 1998–1999 | Twin pregnancies with short CL (≤25 mm) | McDonald | 16–24[ϛ] | No | Gestational age at birth | 27 |

[1]Detailed inclusion criteria not specified.

*Adjusted analyses referred to whether the computation of the risk analyses for the outcomes observed in the present systematic review were adjusted for any factor potentially associated with PTB.

[a]Analysis adjusted for maternal age, pregestational BMI, IVF, operative hysteroscopy, previous cervical surgery, previous spontaneous preterm birth, white blood count, C-reactive protein, neutrophil to lymphocyte ratio and the shortest cervical length at ultrasound.

[b]Not specified on which confounders the analyses were adjusted.

[c]Analyses adjusted for indomethacin, vaginal progesterone, antibiotics and basic demographic characteristics.

[d]Analyses adjusted for cerclage indication, clinical history, age, chorionicity, insurance type, race, BMI, IVF, and multifetal reduction.

[e]Analyses adjusted for age, BMI, race, vaginal progesterone use, and gestational age at shortest documented cervical length.

[f]Analyses adjusted for maternal age, BMI, racial origin, cigarette smoking, IVF, parity, and prior preterm delivery.

[g]Analyses adjusted for amniocentesis and vaginal progesterone administration.

[h]Analyses adjusted for gestational age at presentation and short cervical length.

[§]Standard deviation.

[Ψ]Median and interquartile.

[ϛ]Range.

[Ξ]Mean.

BMI, body mass index; CL, cervical length; DC, dichorionic; DCDA, dichorionic diamniotic; IVF, in vitro fertilization; NR, not reported; PTB, preterm birth; RCT, randomized controlled trial; TVU, transvaginal ultrasound.

included (after removing studies that included overlapping cases) 1,465 twin pregnancies with either short cervical length on ultrasound or cervical dilatation at physical examination. Four studies were randomized and 15 studies were observational.

The results of the quality assessment of the included studies using RoB2 tool are presented in Table 2. The study by Roman and colleagues was at low risk of bias, while those by Rust and colleagues and Althuisius and colleagues were at high risk of bias (Table 2).

The results of the quality assessment of the observational studies are reported in Table 3. Most of the studies were of good quality; the main limitations of the included studies were small sample size, observational design, lack of subgroup analyses according to indication for cerclage, and heterogeneity in the outcomes observed and prenatal management of twin pregnancies undergoing cervical cerclage.

## Synthesis of the results

**Women with short cervical length at ultrasound or cervical dilatation at physical examination.** Placement of cervical cerclage in women with a twin pregnancy with a short cervix at

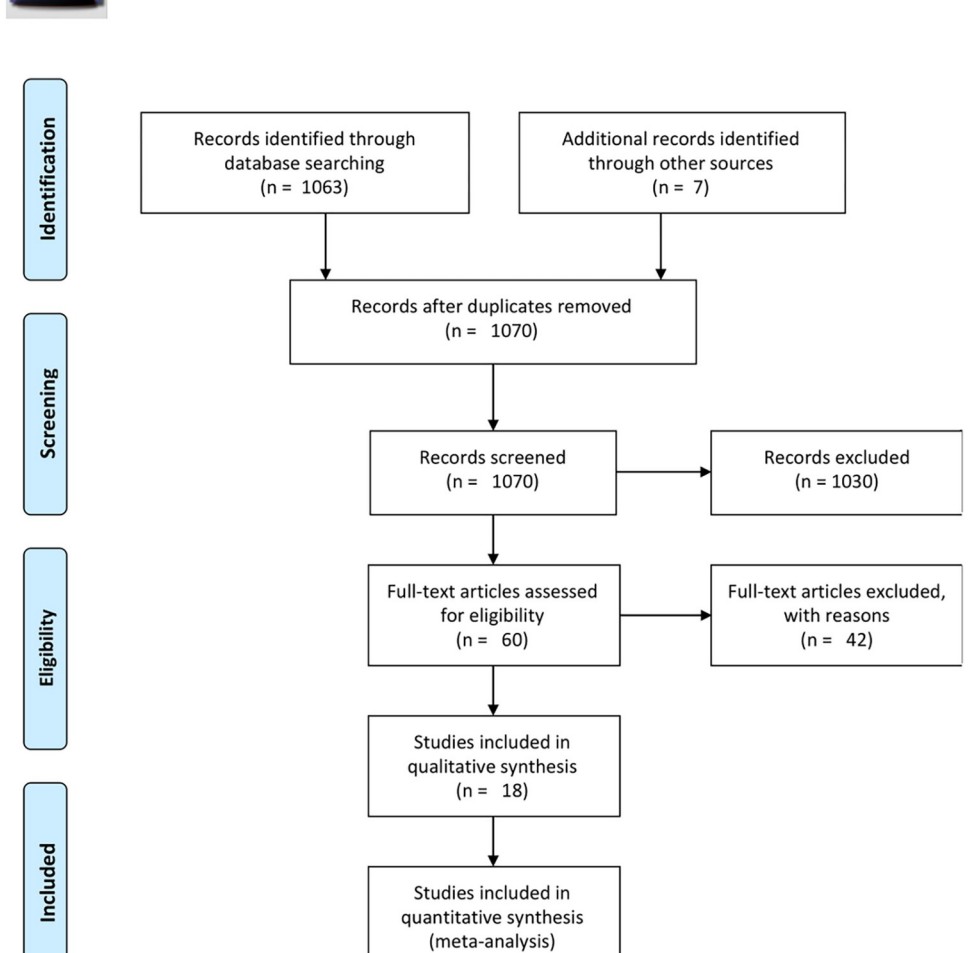

**Fig 1. Systematic review flowchart.**

**Table 2. Risk of bias assessed using the Revised Cochrane risk-of-bias tool for randomized trials "RoB 2".**

| Study ID | Randomization process | Deviations from intended interventions | Missing outcome data | Measurement of the outcome | Selection of the reported result | Overall bias |
|---|---|---|---|---|---|---|
| Roman and colleagues (2020) | Low risk | Low risk | Low risk | Low risk | Low risk | Low risk |
| Althuisius and colleagues (2001) | Low risk | High risk | Low risk | High risk | High risk | High risk |
| Rust and colleagues (2000) | Low risk | High risk | Low risk | High risk | High risk | High risk |

According to this tool, the risk of bias of each included study is judged according to 5 domains: bias arising from the randomization process, bias due to deviations from intended interventions, bias due to missing outcome data, bias in the measurement of the outcome, and bias in selection of the reported result. Although the RoB2 tool does not provide an overall risk of bias assessment, the overall risk of bias was considered low if 4 or more domains were rated as low risk "not counting 'other biases,'" with at least 1 being sequence generation or allocation concealment, according to what is reported in previous systematic reviews of intervention.

**Table 3. Quality assessment of the included studies according to the NOS for cohort studies; a study can be awarded a maximum of one star for each numbered item within the Selection and Outcome categories.** A maximum of 2 stars can be given for Comparability*.

| Author | Year | Selection | Comparability | Outcome |
|---|---|---|---|---|
| Qiu | 2023 | ★★★ | ★★ | ★★ |
| Qiu | 2022 | ★★ | ★ | ★ |
| Yao | 2022 | ★★★ | ★★ | ★★ |
| Zeng | 2022 | ★★ | ★★ | ★★ |
| Pan | 2020 | ★ | ★ | ★ |
| Wu | 2020 | ★★★ | ★★ | ★★ |
| Han | 2020 | ★★ | ★★ | ★★ |
| Qureshey | 2019 | ★★ | ★★ | ★★ |
| Abbasi | 2018 | ★★ | ★ | ★ |
| Adams | 2018 | ★★ | ★★ | ★ |
| Houlihan | 2016 | ★★ | ★★ | ★★ |
| Roman | 2016 | ★★ | ★★ | ★★ |
| Roman | 2015 | ★★★ | ★★ | ★★ |
| Roman | 2005 | ★★★ | ★★ | ★★ |
| Newman | 2002 | ★★ | ★★ | ★★ |

*Higher number of stars indicated a better quality of the study.

NOS, Newcastle–Ottawa scale.

Table 4 reports the main maternal and pregnancy characteristics potentially affecting the risk of PTB in twin pregnancies. There was no significant difference in the mean cervical length at ultrasound [$p = 0.08$] or cervical dilatation at physical examination ($p = 0.05$) between women receiving compared to those not receiving cervical cerclage. Likewise, there was no difference in the mean maternal age ($p = 0.2$), BMI ($p = 0.3$), nulliparity ($p = 0.6$), prior PTB ($n = 0.7$), and pharmacological intervention for reducing the risk of PTB, including indomethacin ($p = 0.11$), antibiotics ($p = 0.4$), and tocolytic drugs ($p = 0.2$) between the 2 groups. Women receiving cerclage were more likely to carry dichorionic gestations (RR: 0.63, 95% CI [0.44, 0.90], $p = 0.01$; 70/691 versus 84/556) and were diagnosed with short cervical length or cervical dilatation at earlier gestational ages compared to those not receiving cerclage (MD: −0.83 weeks, 95% CI [−1.47, −0.19], $p = 0.01$) (Table 4).

**Table 4. Results of the meta-analyses comparing the likelihood of several baseline characteristics [or the mean age] between women undergoing cerclage versus women not undergoing cerclage.**

| Baseline characteristics | Number of studies | n/N vs. n/N | RR [95% CI] | *p* Value | I$^2$ [95% CI], % |
|---|---|---|---|---|---|
| Monochorionic twins | 11 | 70/691 vs. 84/556 | 0.63 [0.44, 0.90] | 0.01 | 16 [0, 56] |
| Nulliparity | 13 | 539/710 vs. 416/565 | 1.09 [0.82, 1.45] | 0.6 | 31 [0, 64] |
| Prior preterm birth | 13 | 61/566 vs. 61/471 | 0.94 [0.64, 1.38] | 0.7 | 0 [0, 57] |
| In vitro fertilization | 11 | 451/640 vs. 289/477 | 1.13 [0.95, 1.34] | 0.2 | 65 [34, 82] |
| Progesterone use | 7 | 242/445 vs. 173/306 | 1.00 [0.99, 1.01] | 0.9 | 0 [0, 71] |
| Indomethacin use | 5 | 148/183 vs. 96/174 | 1.65 [0.90, 3.02] | 0.11 | 99 [99, 100] |
| Antibiotics use | 8 | 245/370 vs. 220/322 | 0.96 [0.88, 1.06] | 0.4 | 86 [74, 92] |
| Steroids use | 6 | 191/267 vs. 104/169 | 1.05 [0.79, 1.41] | 0.7 | 80 [58, 81] |
| Tocolysis | 8 | 245/370 vs. 250/322 | 0.87 [0.71, 1.06] | 0.2 | 98 [97, 98] |
| | | N/N | MD [95% CI] | | |
| Gestational age at diagnosis [weeks] | 15 | 774/627 | −0.83 [−1.47, −0.19] | 0.01 | 77 [62, 86] |
| Maternal age at baseline | 13 | 753/604 | 2.70 [−1.41, 6.81] | 0.2 | 98 [98, 99] |
| Maternal BMI at baseline | 12 | 669/467 | 2.76 [−1.79, 7.31] | 0.3 | 99 [98, 99] |
| Cervical length at baseline | 12 | 620/519 | −0.54 [−0.94, 0.14] | 0.09 | 60 [24, 79] |
| Cervical dilatation at baseline | 7 | 319/216 | −0.58 [−1.16, 0.00] | 0.05 | 91 [83, 95] |

BMI, body mass index; CI, confidence interval; MD, mean difference; RR, risk ratio.

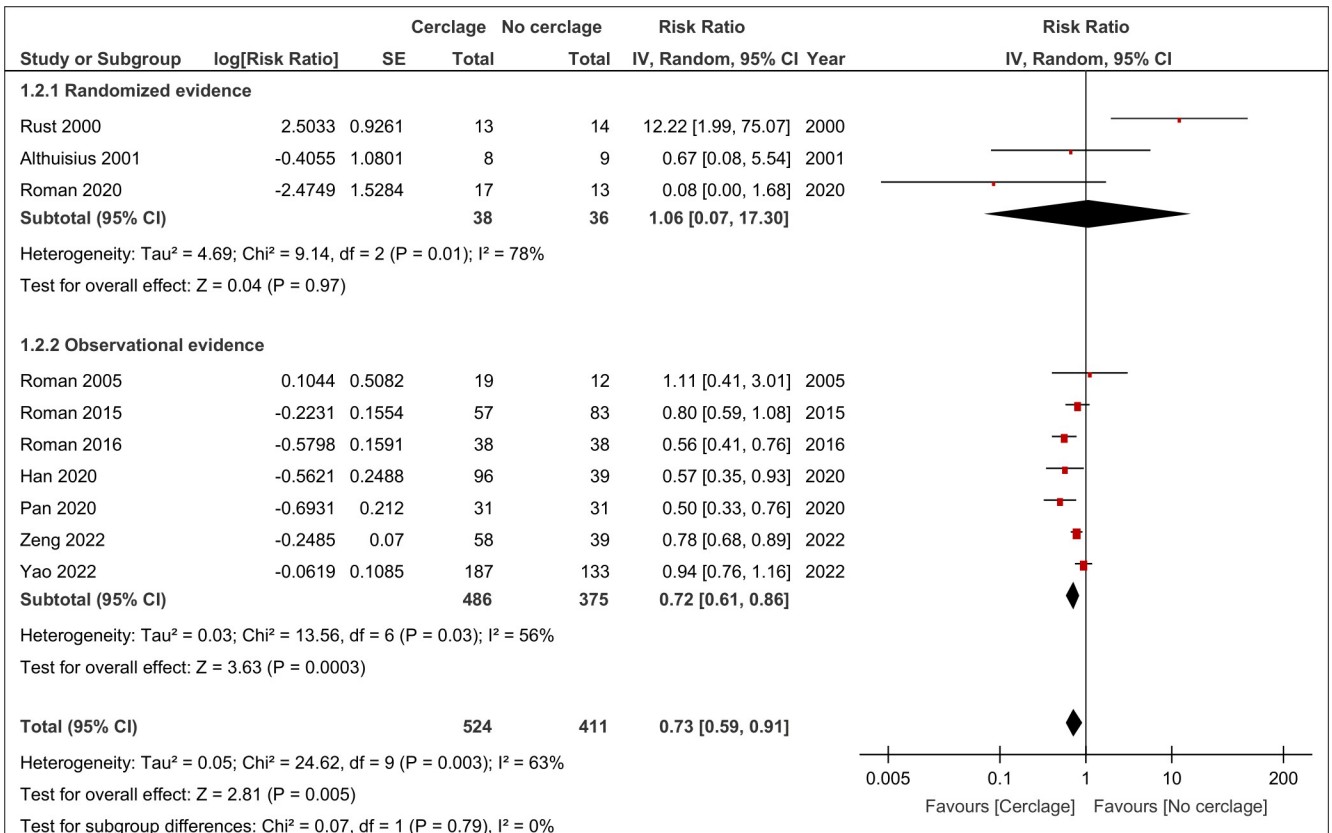

**Fig 2. Pooled ORs for the risk of PTB <34 weeks of gestation in women with twin pregnancies undergoing compared to those not undergoing cervical cerclage.** OR, odds ratio; PTB, preterm birth.

ultrasound or cervical dilatation at physical examination was associated with a reduced risk of PTB <34 weeks of gestation (RR: 0.73, 95% CI [0.59, 0.91], $p = 0.005$, corresponding to a 16% difference in the absolute risk, AR) (Fig 2). The strength of such association was due to the reduced risk of PTB in women with cervical cerclage from the included observational studies (RR: 0.72, 95% CI [0.61, 0.86], $p < 0.001$), but not RCT ($p = 0.9$). Cervical cerclage was also associated with a reduced risk of PTB <32 (RR: 0.69, 95% CI [0.57, 0.84], $p < 0.01$, AR: 16.92%), <28 [RR: 0.54, 95% CI [0.43, 0.67], $p < 0.01$, AR: 18.29%) and <24 (RR: 0.48, 95% CI [0.23, 0.97], $p = 0.04$, AR: 15.57%) but not 37 weeks ($p = 0.2$) of gestation (Table 5). Likewise, cervical cerclage in twin pregnancies with either a short cervical length or cervical dilatation was associated with a prolonged gestational age at birth (MD: 2.32 weeks, 95% CI [0.99, 3.66, $p < 0.001$) and longer presentation to delivery interval (MD: 5.22 weeks, 95% CI [3.86, 6.59], $p < 0.001$) (Table 6).

Conversely, there was no significant difference in the risk of pPROM ($p = 0.2$) or chorioamnionitis ($p = 0.8$) between women receiving and those not receiving cerclage. Pooled proportions for each of the explored outcomes in women with twin pregnancies receiving compared to those not receiving cerclage are reported in S4 Table.

Cerclage in twin pregnancy with short cervical length or cervical dilatation was also associated with a reduced risk of perinatal loss (RR: 0.38, 95% CI [0.25, 0.60], $p < 0.001$ AR: 19.62%), composite adverse outcome (RR: 0.69, 95% CI [0.53, 0.90], $p = 0.007$; AR: 11.75%), 5-min Apgar score <7 (RR: 0.46, 95% CI [0.29, 0.74], $p = 0.00$), neonatal sepsis (RR: 0.45, 95% CI [0.24, 0.84], $p = 0.01$), grade III or IV IVH (RR: 0.32, 95% [CI 0.11, 0.92], $p = 0.03$), birthweight

**Table 5. Women with a reduced cervical length on ultrasound and/or cervical dilatation at examination: Results of the head-to-head meta-analyses comparing the risk of selected categorical outcomes in women with twin pregnancies undergoing cerclage versus no cerclage.**

| Outcomes [Ref.] | Number of studies | Total women n/N vs. n/N | RR [95% CI] | p Value | I² [95% CI], % |
|---|---|---|---|---|---|
| **Primary outcome:** | | | | | |
| **Preterm birth <34th week** | **10** | **258/524 vs. 268/411** | **0.73 [0.59, 0.91]** | **0.005** | **63 [28, 81]** |
| - Randomized evidence | 3 | 24/38 vs. 19/36 | 1.06 [0.07,17.3] | 0.9 | 78 [29, 93] |
| - Observational evidence | 7 | 234/486 vs. 249/375 | 0.72 [0.61, 0.86] | <0.001 | 56 [0, 81] |
| **Preterm birth <37th week** | **6** | **275/404 vs. 247/317** | **0.95 [0.87, 1.03]** | **0.2** | **3 [0, 75]** |
| - Randomized evidence | 2 | 19/21 vs. 17/23 | 3.25 [0.57,18.7] | 0.2 | 0 [--] |
| - Observational evidence | 4 | 256/383 vs. 230/394 | 0.94 [0.87,1.02] | 0.2 | 1 [0, 85] |
| **Preterm birth <32nd week** | **12** | **239/619 vs. 276/497** | **0.69 [0.57, 0.84]** | **<0.001** | **64 [27, 83]** |
| - Randomized evidence | 3 | 19/38 vs. 16/36 | 1.28 [0.36, 4.54] | 0.7 | 66 [0, 90] |
| - Observational evidence | 9 | 220/581 vs. 260/461 | 0.68 [0.55, 0.82] | <0.001 | 64 [26, 82] |
| **Preterm birth <28th week** | **11** | **119/523 vs. 188/458** | **0.54 [0.43, 0.67]** | **<0.001** | **29 [0, 65]** |
| - Randomized evidence | 3 | 12/38 vs. 12/36 | 1.35 [0.25, 7.14] | 0.3 | 61 [0, 89] |
| - Observational evidence | 8 | 107/485 vs. 176/422 | 0.52 [0.43, 0.64] | <0.001 | 19 [0, 62] |
| **Preterm birth <24th week** | **7** | **29/222 vs. 65/227** | **0.48 [0.23,0.97]** | **0.04** | **62 [14, 83]** |
| - Randomized evidence | 3 | 7/38 vs. 11/36 | 0.77 [0.17,3.54] | 0.7 | 45 [0, 84] |
| - Observational evidence | 4 | 22/184 vs. 54/191 | 0.42 [0.116,1.11] | 0.08 | 75 [30, 91] |
| **pPROM** | **8** | **105/404 vs. 126/324** | **0.75 [0.48, 1.16]** | **0.2** | **68 [32, 85]** |
| - Randomized evidence | 3 | 16/38 vs. 9/36 | 1.57 [0.81, 3.04] | 0.2 | 0 [0, 90] |
| - Observational evidence | 5 | 89/366 vs. 117/288 | 0.60 [0.39, 0.92] | 0.02 | 66 [12, 87] |
| **Chorioamnionitis** | **7** | **37/409 vs. 26/270** | **1.08 [0.54, 2.17]** | **0.8** | **50 [0, 79]** |
| - Randomized evidence | 3 | 12/38 vs. 13/36 | 1.03 [0.23, 4.65] | 0.9 | 70 [0, 91] |
| - Observational evidence | 4 | 25/371 vs. 13/234 | 1.24 [0.70, 2.21] | 0.5 | 0 [0, 85] |
| **Perinatal loss** | **9** | **131/980 vs. 223/676*** | **0.38 [0.25, 0.60]** | **<0.001** | **72 [44, 86]** |
| - Randomized evidence | 3 | 9/76 vs. 22/54 | 0.43 [0.04, 4.63] | 0.5 | 79 [32, 93] |
| - Observational evidence | 6 | 122/904 vs. 201/622 | 0.42 [0.28, 0.63] | <0.001 | 71 [31, 87] |
| **Composite adverse outcome** | **8** | **418/904 vs. 283/488** | **0.69 [0.53, 0.90]** | **0.007** | **83 [69, 91]** |
| - Randomized evidence | 1 | 14/30 vs. 3/6 | 0.93 [0.38, 2.27] | 0.9 | ,, |
| - Observational evidence | 7 | 404/874 vs. 280/482 | 0.67 [0.50, 0.90] | 0.007 | 86 [73,93] |
| **5-min Apgar score <7** | **5** | **97/346 vs. 126/212** | **0.46 [0.29, 0.74]** | **0.001** | **75 [38,90]** |
| - Randomized evidence | 1 | 9/34 vs. 22/26 | 0.31 [0.17, 0.56] | <0.001 | – |
| - Observational evidence | 4 | 88/312 vs. 104/186 | 0.50 [0.29, 0.89] | 0.02 | 79 [42,92] |
| **RDS** | **4** | **70/224 vs. 56/160** | **1.13 [0.49, 2.62]** | **0.8** | **80 [47,92]** |
| - Randomized evidence | 1 | 14/30 vs. 2/6 | 1.40 [0.42, 4.62] | 0.6 | – |
| - Observational evidence | 3 | 56/194 vs. 54/154 | 1.09 [0.40, 2.97] | 0.9 | 85 [54, 95] |
| **Sepsis** | **3** | **14/138 vs. 20/84** | **0.45 [0.24, 0.84]** | **0.01** | **0 [0, 90]** |
| - Randomized evidence | 1 | 2/30 vs. 1/6 | 0.40 [0.04, 3.74] | 0.4 | – |
| - Observational evidence | 2 | 12/108 vs. 19/78 | 0.46 [0.24, 0.87] | 0.6 | 0 [--] |
| **Grade 3–4 IVH** | **4** | **16/224 vs. 42/160** | **0.32 [0.11, 0.92]** | **0.03** | **56 [0,85]** |
| - Randomized evidence | 1 | 4/30 vs. 1/6 | 0.80 [0.11, 5.96] | 0.8 | – |
| - Observational evidence | 3 | 12/194 vs. 41/154 | 0.26 [0.08, -0.90] | 0.03 | 63 [0, 89] |
| **ROP** | **4** | **14/224 vs. 17/160** | **0.54 [0.10, 2.98]** | **0.2** | **66 [0, 88]** |
| - Randomized evidence | 1 | 5/30 vs. 1/6 | 1.00 [0.10, 10.5] | 0.99 | – |
| - Observational evidence | 3 | 9/194 vs. 16/154 | 0.46 [0.05, 4.31] | 0.50 | 76 [20, 93] |
| **Birthweight <1,500 g** | **5** | **228/638 vs. 264/454** | **0.49 [0.33, 0.73]** | **<0.001** | **85 [66, 93]** |
| - Randomized evidence | 1 | 21/34 vs. 24/26 | 0.13 [0.03, 0.67] | 0.01 | – |
| - Observational evidence | 4 | 207/604 vs. 240/428 | 0.53 [0.36, 0.78] | 0.001 | 87 [69, 95] |

*(Continued)*

**Table 5.** (Continued)

| Outcomes [Ref.] | Number of studies | Total women n/N vs. n/N | RR [95% CI] | p Value | I² [95% CI], % |
|---|---|---|---|---|---|
| **NICU admission** | **7** | **452/822 vs. 306/464** | **0.75 [0.63, 0.90]** | **<0.001** | **74 [46, 88]** |
| - Randomized evidence | 1 | 22/30 vs. 6/6 | 0.20 [0.01, 4.02] | 0.3 | – |
| - Observational evidence | 6 | 430/792 vs. 300/458 | 0.76 [0.63, 0.91] | 0.003 | 78 [51, 90] |

*N. of fetuses.

RR, risk ratio; CI, confidence interval; n/N vs. n/N, number of women with the outcome/total number of women in the exposed [cerclage] and unexposed [no cerclage] group, respectively; PTB, preterm birth; pPROM, preterm premature rupture of membranes; RDS, respiratory distress syndrome; ROP, retinopathy of the prematurity; IVH, intraventricular hemorrhage; NICU, neonatal intensive care unit.

<1,500 grams (RR: 0.49, 95% CI [0.33, 0.73], $p < 0.01$), and NICU admission (RR: 0.75, 95% CI [0.63, 0.90], $p < 0.001$] but not of RDS ($p = 0.8$) or ROP ($p = 0.2$).

Mean birthweight was also greater in twin pregnancies receiving cerclage [MD: 300 grams, 95% CI 167, 433; $p < 0.01$], while the length of stay in NICU was shorter [MD: −22.4 days, 95% CI −40.1, −4.7; $p = 0.01$]. When assessing the contribution of the different types of studies included in the reported results, the association between cervical cerclage and adverse maternal or perinatal outcome was exclusively due to the inclusion of observational studies but not RCTs.

Subgroup analyses according to the specific indication for cerclage (short cervical length at ultrasound or cervical dilation at physical examination) are presented in Tables 6–8.

In women with a CL ≤15 mm, placement of a cervical cerclage was associated with a reduced risk of PTB <34 weeks (RR: 0.74, 95% CI [0.58, 0.95], $p < 0.001$, AR: 29.17%) and composite adverse neonatal outcome (RR: 0.75, 95% CI [0.61, 0.92; 0.03], $p = 0.003$, AR: 22.64%) (Table 7). Cerclage was also associated with a later gestational age at birth (MD: 2.34, 95% CI 1.40, 3.28, $p < 0.001$) and a longer presentation to delivery interval (MD: 3.79, 95% CI [2.42, 5.15], $p < 0.001$) and neonatal birthweight (MD: 627 grams, 95% CI [57.6, 1,196], $p = 0.003$). The association between cerclage and reduced risk of maternal and perinatal outcome was due to the inclusion of observational studies, while the RCT did not show any potential beneficial effect of cerclage in affecting such outcomes.

Conversely, cerclage in women with a cervical length between 15 and 25 mm was not associated with a reduced risk of any of the main outcomes assessed in this systematic review.

In women with twin pregnancy and cervical dilatation at physical examination, placement of a cervical cerclage was associated with a reduced risk of PTB <34 (RR: 0.68, 95% CI [0.57, 0.80], $p = 0.001$), <32 (RR: 0.59, 95% CI [0.50, 0.70], $p = 0.001$), <28 (RR: 0,47 95% CI [0.36, 0.62], $p < 0.001$), and <24 weeks (RR: 0.32 95% CI [0.21, 0.48], $p < 0.001$), but not that of pPROM ($p = 0.3$), chorioamnionitis ($p = 0.5$). Cerclage in these women also reduced the risk of perinatal loss (RR: 0.30, 95% CI [0.16, 0.55], $p < 0.001$), Apgar score <7 at 5 min (RR: 0.49, 95% CI [0.27, 0.90], $p < 0.001$), birthweight <1,500 grams (RR: 0.41, 95% CI [0.31, 0.55], $p < 0.001$), and admission to NICU (RR: 0.67, 95% CI [0.52, 0.88], $p = 0.003$), but not that of RDS ($p = 0.5$), grades III and IV IVH ($p = 0.2$) or ROP ($p = 0.3$) (Table 8). Such association was due to the inclusion of observational studies but no RCTs. Unfortunately, we could not perform meaningful pooled subgroup analyses according to different degrees of cervical dilatation [>2, >3, >4 cm]. Likewise, we could not perform sub-analyses according to chorionicity.

## Grade

Assessment of the quality of retrieved evidence according to GRADE is presented in S5 Table. Overall, a low quality of evidence showed that cerclage can reduce the risk of PTB <34 weeks

**Table 6. Results of the meta-analyses comparing selected continuous perinatal outcomes in women with twin pregnancies undergoing cerclage versus no cerclage.**

| Outcomes | Number of studies [total sample] | MD [95% CI] | p Value | $I^2$ [95% CI], % |
|---|---|---|---|---|
| *1. Cerclage for reduced cervical length on ultrasound or cervical dilatation at physical examination* | | | | |
| **Gestational age at birth, (weeks)** | **17 [1,426]** | **2.32 [0.99, 3.66]** | **<0.001** | **86 [78, 90]** |
| - Randomized evidence | 2 [48] | −0.09 [−0.26, 2.07] | 0.9 | [−−] |
| - Observational evidence | 15 [1,378] | 2.54 [1.13, 3.95] | <0.001 | 87 [80, 91] |
| **Presentation to delivery interval, (weeks)** | **11 [801]** | **5.22 [3.86, 6.59]** | **<0.001** | **90 [84, 94]** |
| - Randomized evidence | 1 [30] | 5.40 [2.20, 8.60] | 0.001 | – |
| - Observational evidence | 10 [771] | 5.21 [3.77, 6.65] | <0.001 | 91 [86, 94] |
| **Birthweight, grams** | **15 [1,483]** | **300 [167, 433]** | **<0.001** | **90 [85, 93]** |
| - Randomized evidence | 1 [60] | 268 [132, 403] | <0.001 | – |
| - Observational evidence | 14 [1,423] | 805 [467, 1143] | <0.001 | 90 [84, 93] |
| **NICU length of stay, [days]** | **6 [702]** | **−22.4 [−40.1, −4.7]** | **0.01** | **93 [87 96]** |
| - Randomized evidence | 1 [56] | −22.2 [−41.4, −3.0] | 0.02 | – |
| - Observational evidence | 5 [646] | −24.1 [−55.2, 7.0] | 0.13 | 94 [89, 97] |
| *2. Cerclage for cervical dilatation at physical examination:* | | | | |
| **Gestational age at birth, (weeks)** | **5 [345]** | **3.64 [1.85, 5.43]** | **<0.001** | **68 [17, 88]** |
| - Randomized evidence | 1 [30] | 0.0 [−8.54, 8.54] | 0.99 | – |
| - Observational evidence | 4 [314] | 3.79 [1.92, 5.65] | <0.001 | 75 [30, 91] |
| **Presentation to delivery interval, (weeks)** | **5 [334]** | **5.43 [3.28, 7.57]** | **<0.001** | **95 [92, 97]** |
| - Randomized evidence | 1 [30] | 5.40 [2.20, 8.60] | 0.01 | – |
| - Observational evidence | 4 [304] | 5.43; [3.04, 7.81] | <0.001 | 97 [94, 98] |
| **Birthweight, grams** | **5 [375]** | **500 [297, 703]** | **<0.001** | **79 [51, 91]** |
| - Randomized evidence | 1 [60] | 805 [468, 1143] | <0.001 | – |
| - Observational evidence | 4 [315] | 442 [230, 654] | <0.001 | 80 [46, 92] |
| **NICU length of stay, (days)** | **3 [203]** | **−36.7 [−56.4, −17.0]** | **<0.001** | **64 [0, 90]** |
| - Randomized evidence | 1 [56] | −24.1 [−55.2, 7.0] | <0.001 | – |
| - Observational evidence | 2 [147] | −40.8 [−67.2, −14.4] | <0.001 | 79 [−−] |
| *3. Cerclage for reduced cervical length on ultrasound [<25mm]:* | | | | |
| **Gestational age at birth, (weeks)** | **10 [989]** | **1.02 [−0.43, 2.46]** | **0.2** | **79 [62, 88]** |
| - Randomized evidence | 1 [17] | −0.10 [−2.34, 2.14] | 0.2 | – |
| - Observational evidence | 9 [972] | 1.16 [−0.44, 2.76] | 0.9 | 81 [65, 90] |
| **Presentation to delivery interval, (weeks)** | | | | |
| - Observational evidence only | 5 [346] | 5.20 [2.29, 8.11] | <0.001 | 88 [74, 94] |
| **Birthweight, grams** | | | | |
| - Observational evidence only | 8 [911] | 183 [−9.9, 376] | 0.06 | 836 [68, 91] |
| *4. Cerclage for reduced cervical length on ultrasound [stratified by cervical length]:* | | | | |
| **Gestational age at birth, (weeks)- <15 mm** | | | | |
| - Observational evidence only | 5 [366] | 2.34 [1.40, 3.28] | <0.001 | 0 [0, 79] |
| **Gestational age at birth, (weeks)- 15–25 mm** | | | | |
| - Observational evidence only | 4 [278] | 1.36 [−1.26, 3.97] | 0.3 | 57 [0, 89] |
| **Presentation to delivery interval, (weeks)—<15 mm** | | | | |
| - Observational evidence only | 4 [195] | 3.79 [2.42, 5.15] | <0.001 | 0 [0, 85] |
| **Presentation to delivery interval, (weeks)- 15–25 mm** | | | | |
| - Observational evidence only | 3 [123] | 3.00 [0.91, 5.08] | 0.05 | 25 [0, 92] |
| **Birthweight, grams- <15 mm** | | | | |
| - Observational evidence only | 3 [544] | 627 [57.6, 1,196] | 0.003 | 98 [96, 99] |
| **Birthweight, grams- 15–25 mm** | | | | |

(*Continued*)

**Table 6.** (Continued)

| Outcomes | Number of studies [total sample] | MD [95% CI] | p Value | I² [95% CI], % |
|---|---|---|---|---|
| - Observational evidence only | 2 [382] | 78.1 [−3.76, 533] | 0.7 | 85 [−−] |

CI, confidence interval; MD, mean difference; NICU, neonatal intensive care unit.

of gestation in women with a short cervix at ultrasound or cervical dilatation at physical examination and this could be due to the considerable inclusion of observational studies, indirectness of evidence, imprecision of results, and publication bias.

## Discussion

The findings from this systematic review showed that there is still a low grade of evidence that cerclage may prevent PTB in twin pregnancies. Although the placement of cervical cerclage in women with short cervical length <15 mm or cervical dilatation may be potentially associated with a reduced risk of PTB and adverse perinatal outcome compared with no intervention, this evidence is mainly supported by observational studies, but no RCTs, although only 1 trial was published in the last few years.

This is, to the best of our knowledge, the largest and most up-to-date systematic review and meta-analysis on the role of cervical cerclage in affecting PTB in twin pregnancies. Previous systematic reviews have addressed the association between cerclage and perinatal outcome in twins [12,48–51]. Compared to this review, the present study includes a well-defined population of twin pregnancies at high risk of PTB, defined as the presence of a short cervical length at ultrasound or cervical dilatation at physical examination, a large number of outcomes explored, stratification of the analyses according to cervical length at ultrasound or cervical dilatation, and computation of the observed outcome according to the study design (observational versus RCT).

The small number of cases in some of the included studies, their nonrandomized design, lack of standardized criteria for prenatal assessment, and management of twin pregnancies at higher risk of PTB represent the main limitation of the present review. The most significant limitation of the present systematic review relies on the inclusion of mainly observational studies. The large majority of RCTs were old, with a very small number of cases and a heterogeneous population of twin pregnancies, thus considerably limiting the robustness of their findings. Only 1 RCT was published in the recent past, showing a potential beneficial role of cerclage in women with cervical dilatation. However, even this trial, despite being powered for its primary outcome, was limited by a very small number of included cases and also by potential confounders such as the use of indomethacin and antibiotics in the intervention arm. The assessment of the role of cerclage in twin pregnancies with different cut-offs of cervical length was limited by the small number of included cases and an even smaller number of events that might have precluded a robust assessment of the strength of association between cerclage placement and neonatal morbidity in twins.

PTB is the leading cause of perinatal mortality and morbidity worldwide with an estimated societal economic burden in the United States of $26.2 billion annually. Therefore, identifying pregnancies at higher risk of PTB is pivotal in applying preventive strategies. In singleton pregnancies, assessment of cervical length at mid-gestation allows the identification of women with a higher likelihood of delivering preterm. Several preventive strategies for PTB in singleton pregnancies have been proposed. A recent network meta-analysis comparing progesterone, pessary, or cerclage for the prevention of PTB in singleton pregnancies has reported that

**Table 8. Women with cervical dilatation at physical examination: Results of the head-to-head meta-analyses comparing the risk of selected categorical outcomes in women with twin pregnancies undergoing cerclage versus no cerclage.**

| Outcomes | Number of studies | Total women n/N vs. n/N | RR [95% CI] | p Value | I² [95% CI], % |
|---|---|---|---|---|---|
| *Primary outcome:* | | | | | |
| **Preterm birth <34th week** | **5** | **111/194 vs. 107/116** | **0.68 [0.57, 0.80]** | **<0.001** | **43 [0, 78]** |
| - Randomized evidence | 1 | 12/17 vs. 13/13 | 0.08 [0.00, 1.68] | 0.11 | – |
| - Observational evidence | 4 | 99/177 vs. 94/103 | 0.66 [0.53, 0.82] | 0.002 | 60 [0, 84] |
| **Preterm birth <32nd week** | **6** | **130/246 vs. 147/163** | **0.59 [0.50, 0.70]** | **<0.001** | **19 [0, 70]** |
| - Randomized evidence | 1 | 11/17 vs. 13/13 | 0.66 [0.46, 0.95] | 0.03 | – |
| - Observational evidence | 5 | 119/229 vs. 130/154 | 0.61 [0.44, 0.83] | 0.002 | 85 [59, 90] |
| **Preterm birth <28th week** | **5** | **82/192 vs. 126/146** | **0.47 [0.36, 0.62]** | **<0.001** | **39[0, 79]** |
| - Randomized evidence | 1 | 7/17 vs. 11/13 | 0.49 [0.26, 0.90] | 0.02 | – |
| - Observational evidence | 4 | 75/175 vs. 115/133 | 0.50 [0.39, 0.66] | <0.001 | 53 [0, 83] |
| **Preterm birth <24th week** | **4** | **25/140 vs. 56/99** | **0.32 [0.21, 0.48]** | **<0.001** | **0 [0, 68]** |
| - Randomized evidence | 1 | 5/17 vs. 11/13 | 0.35 [0.16, 0.75] | 0.007 | – |
| - Observational evidence | 3 | 20/123 vs. 45/86 | 0.31 [0.18, 0.42] | <0.001 | 16 [0, 77] |
| **pPROM** | **4** | **62/192 vs. 71/146** | **0.68 [0.33, 1.40]** | **0.3** | **80 [18, 91]** |
| - Randomized evidence | 1 | 11/17 vs. 5/13 | 1.78 [0.68, 3.74] | 0.2 | – |
| - Observational evidence | 4 | 51/175 vs. 66/133 | 0.62 [0.33, 1.14] | 0.07 | 74 [0, 89] |
| **Chorioamnionitis** | **3** | **17/102 vs. 12/61** | **1.95 [0.32, 11.7]** | **0.5** | **93 [85, 97]** |
| - Randomized evidence | 1 | 6/17 vs. 11/13 | 0.42 [0.21, 0.83] | <0.001 | |
| - Observational evidence | 3 | 11/85 vs. 1/48 | 2.90 [0.56, 14.98] | 0.203 | 0 [--] |
| **Perinatal loss \*** | **3** | **59/226 vs. 129/180** | **0.30 [0.16, 0.55]** | **<0.001** | **77 [26, 93]** |
| - Randomized evidence | 1 | 6/34 vs. 20/26 | 0.06 [0.02, 0.23] | <0.001 | – |
| - Observational evidence | 2 | 53/192 vs. 109/154 | 0.39 [0.28, 0.53] | <0.001 | 34 [--] |
| **Composite adverse outcome** | **4** | **114/258 vs. 72/116** | **0.64 [0.39, 1.04]** | **0.07** | **79 [42, 92]** |
| - Randomized evidence | 1 | 14/30 vs. 3/26 | 0.93 [0.38, 2.27] | 0.9 | – |
| - Observational evidence | 3 | 100/228 vs. 69/90 | 0.59 [0.34, 1.03] | 0.07 | 84 [52, 95] |
| **5-min Apgar score <7** | **4** | **84/280 vs. 89/150** | **0.49 [0.27, 0.90]** | **0.02** | **80 [46, 92]** |
| - Randomized evidence | 1 | 9/30 vs. 22/26 | 0.31 [0.17, 0.56] | <0.001 | – |
| - Observational evidence | 3 | 75/250 vs. 67/124 | 0.59 [0.27, 1.21] | 0.14 | 83 [50, 95] |
| **RDS** | **2** | **39/84 vs. 41/68** | **0.71 [0.27, 1.83]** | **0.5** | **63 [--]** |
| - Randomized evidence | 1 | 14/30 vs. 2/26 | 0.40 [0.42, 4.62] | 0.6 | – |
| - Observational evidence | 1 | 25/54 vs. 39/42 | 0.50 [0.37, 0.68] | <0.001 | – |
| **Sepsis** | **2** | **9/84 vs. 11/68** | **0.52 [0.23, 1.15]** | **0.11** | **0 [--]** |
| - Randomized evidence | 1 | 2/30 vs. 1/26 | 0.40 [0.04 3.74] | 0.4 | – |
| - Observational evidence | 1 | 7/54 vs. 10/42 | 0.54 [0.23, 1.27] | 0.2 | – |
| **Grades 3–4 IVH** | **2** | **6/84 vs. 18/68** | **0.24 [0.03, 2.03]** | **0.2** | **66 [--]** |
| - Randomized evidence | 1 | 4/30 vs. 1/26 | 0.80 [0.11, 5.96] | 0.8 | – |
| - Observational evidence | 1 | 2/54 vs. 17/42 | 0.09 [0.02, 2.41] | 0.002 | – |
| **ROP** | **2** | **6/84 vs. 10/68** | **0.29 [0.03, 3.05]** | **0.3** | **53 [--]** |
| - Randomized evidence | 1 | 5/30 vs. 1/26 | 1.00 [0.10, 10.5] | 0.99 | – |
| - Observational evidence | 1 | 1/54 vs. 9/42 | 0.09 [0.01, 0.81] | 0.03 | – |
| **Birthweight <1,500 g** | **3** | **84/202 vs. 115/126** | **0.41 [0.31, 0.55]** | **<0.001** | **32 [0, 93]** |
| - Randomized evidence | 1 | 31/34 vs. 24/26 | 0.13 [0.03, 0.67] | 0.001 | – |
| - Observational evidence | 2 | 63/168 vs. 91/100 | 0.43 [0.35, 0.54] | <0.001 | 0 [--] |
| **NICU admission** | **3** | **117/176 vs. 71/92** | **0.67 [0.52, 0.88]** | **0.003** | **63 [0, 89]** |
| - Randomized evidence | 1 | 22/30 vs. 6/26 | 0.20 [0.01, 4.02] | 0.3 | – |

*(Continued)*

**Table 8.** (Continued)

| Outcomes | Number of studies | Total women n/N vs. n/N | RR [95% CI] | p Value | I² [95% CI], % |
|---|---|---|---|---|---|
| - Observational evidence | 2 | 95/168 vs. 65/100 | 0.68 [0.52, 0.89] | 0.006 | 79 [--] |

*N. of fetuses.

RR, risk ratio; CI, confidence interval; n/N vs. n/N, number of women with the outcome/total number of women in the exposed [cerclage] and unexposed [no cerclage] group, respectively; pPROM, preterm premature rupture of membranes; RDS, respiratory distress syndrome; ROP, retinopathy of the prematurity; IVH, intraventricular hemorrhage; NICU, neonatal intensive care unit.

vaginal progesterone in pregnancies at high risk was the only intervention with consistent effectiveness and was associated with a significant reduction in the risk of PTB <34 and <37 weeks' gestation and in the risk of neonatal death [10]. Placement of cervical cerclage is commonly considered a secondary preventive strategy for PTB, especially in asymptomatic women with reduced cervical length already taking progesterone therapy. A recent individual patient data (IPD) meta-analysis comparing insertion of cerclage with expectant management reported no significant reduction in PTB <35 weeks' gestation in asymptomatic women with a singleton pregnancy and a short second trimester cervical length (<25 mm). However, a sub-group analysis of the same cohort including women with cervical length <10 mm demonstrated a significant reduction in PTB <35 weeks [52]. On this basis, most relevant national and international societies suggest follow-up ultrasound scans every 1 to 2 weeks up to 24 weeks' gestation in women with reduced cervical length and recommend cerclage placement in those whose cervix shortens to <10 mm despite using progesterone [53].

Screening for PTB in twin pregnancies is more controversial. The International Society of Ultrasound in Obstetrics and Gynecology (ISUOG) recommends that cervical length should be assessed in both monochorionic and dichorionic twin pregnancies at 20 weeks of gestation [54]. However, although in asymptomatic women with twin pregnancies, a short cervical length at ultrasound is associated with a higher risk of PTB, the diagnostic performance of this test is lower than in singletons [55,56]. Furthermore, the optimal cut-off of cervical length to define a twin pregnancy at increased risk of PTB remains controversial. Conventionally, a cut-off of 25 mm, as in singletons, is used.

The effectiveness of the most common strategies for the prevention of PTB is also controversial. Bed rest, progesterone therapy, Arabin cervical pessary, or oral tocolytics do not reduce the risk of PTB in twin pregnancies. A recent network meta-analysis reported that cervical pessary, progesterone, and cerclage do not show a significant effect in reducing the rate of PTB or perinatal morbidity in twins, either in an unselected population of twins or in pregnancies with a short cervix [12]. However, in this review, only 3 small RCTs on cerclage in twins were included. These studies were published almost 2 decades ago and were limited by the very small number of included cases and an even smaller number of events, as well as large heterogeneity in the prenatal management of twin pregnancies with risk factors for PTB, thus preventing robust conclusions on the lack of effectiveness of cerclage in twin pregnancies. More recently, an RCT by Roman and colleagues has reported that, in asymptomatic twin pregnancies with cervical dilation of 1 to 5 cm between $16^{+0}$ and $23^{+6}$ weeks of gestation, placement of cervical cerclage was associated with a significant reduction of PTB <34, 32, 28, and 24 weeks of gestation and a higher mean gestational age at birth (29.05 ± 1.7 versus 22.5 ± 3.9 weeks). Perinatal mortality was also significantly reduced in the cerclage group compared with the no cerclage group [15]. Since the publication of this trial, many observational studies on the role of cerclage in twin pregnancies have been published, challenging the old dogma of its lack of effectiveness in preventing PTB.

**Table 7. Women with a reduced cervical length on ultrasound: Results of the head-to-head meta-analyses comparing the risk of selected categorical outcomes in women with twin pregnancies undergoing cerclage versus no cerclage.**

| Outcomes | Number of studies | Total women n/N vs. n/N | RR [95% CI] | p Value | I² [95% CI], % |
|---|---|---|---|---|---|
| *Primary outcome*: | | | | | |
| **Preterm birth <34th week** | **5** | **143/248 vs. 135/251** | **0.99 [0.68, 1.45]** | **0.9** | **55 [0, 83]** |
| - Randomized evidence | 2 | 12/21 vs. 6/23 | 3.01 [0.17, 51.9] | 0.5 | 0 [--] |
| - Observational evidence | 3 | 131/263 vs. 129/228 | 0.90 [0.76, 1.07] | 0.2 | 0 [0, 90] |
| *By cervical length*: | | | | | |
| **Preterm birth <34th week—<15mm** | | | | | |
| - Observational evidence only | 2 | 29/56 vs. 51/63 | 0.74 [0.58, 0.95] | 0.02 | 0 [--] |
| **Preterm birth <34th week—15–25mm** | | | | | |
| - Observational evidence only | 1 | 11/21 vs. 17/21 | 0.65 [0.41, 1.02] | 0.07 | – |
| **Preterm birth <37th week** | **5** | **237/308 vs. 223/278** | **0.96 [0.88, 1.04]** | **0.3** | **0 [0, 79]** |
| - Randomized evidence | 2 | 19/21 vs. 17/23 | 3.25 [0.57, 18.7] | 0.2 | 0 [--] |
| - Observational evidence | 3 | 218/287 206/255 | 0.96 [0.88, 1.04] | 0.2 | 0 [0, 90] |
| **Preterm birth <32nd week** | **6** | **110/327 vs. 108/290** | **0.90 [0.71, 1.13]** | **0.9** | **5 [0, 76]** |
| - Randomized evidence | 2 | 8/21 vs. 3/23 | 2.89 [0.86, 9.78] | 0.09 | 0 [--] |
| - Observational evidence | 4 | 102/306 vs. 105/267 | 0.86 [0.70, 1.07] | 0.2 | 5 [0, 76] |
| **Preterm birth <28th week** | **6** | **46/327 vs. 55/290** | **0.75 [0.53, 1.08]** | **0.13** | **0 [0, 75]** |
| - Randomized evidence | 2 | 5/21 vs. 1/23 | 3.98 [0.72, 22.0] | 0.11 | 0 [--] |
| - Observational evidence | 4 | 41/306 vs. 54/267 | 0.70 [0.48, 1.01] | 0.06 | 0 [0, 85] |
| **Preterm birth <24th week** | **3** | **9/78 vs. 5/106** | **2.08 [0.80, 5.39]** | **0.13** | **0 [0, 90]** |
| - Randomized evidence | 2 | 2/21 vs. 0/23 | 2.23 [0.32, 15.7] | 0.4 | 0 [--] |
| - Observational evidence | 1 | 7/57 vs. 5/83 | 2.03 [0.68, 6.06] | 0.2 | – |
| **pPROM** | **3** | **46/208 vs. 47/156** | **0.76 [0.44, 1.32]** | **0.3** | **12 [0, 91]** |
| - Randomized evidence | 2 | 5/21 vs. 4/23 | 1.36 [0.31, 6.07] | 0.7 | 0 [--] |
| - Observational evidence | 1 | 41/187 vs. 43/133 | 0.68 [0.47, 0.98] | 0.04 | – |
| **Chorioamnionitis** | **3** | **8/208 vs. 3/156** | **2.29 [0.76, 6.93]** | **0.14** | **0 [0, 90]** |
| - Randomized evidence | 2 | 6/21 vs. 2/23 | 2.54 [0.74, 8.79] | 0.7 | 0 [--] |
| - Observational evidence | 1 | 2/187 vs. 1/133 | 1.52 [0.13, 17.8] | 0.14 | – |
| **Perinatal loss** | **4** | **61/502 vs. 60/372*** | **0.77 [0.55, 1.07]** | **0.12** | **0 [0, 85]** |
| - Randomized evidence | 2 | 3/42 vs. 2/28 | 0.50 [0.31, 7.20] | 0.6 | 0 [--] |
| - Observational evidence | 2 | 58/460 vs. 58/344 | 0.74 [0.52, 1.05] | 0.09 | 1 [--] |
| **Composite adverse outcome** | **4** | **185/449 vs. 144/340** | **1.11 [0.63, 1.96]** | **0.7** | **68 [7, 89]** |
| - Randomized evidence | 2 | 18/42 vs. 12/46 | 0.87 [0.74, 1.02] | 0.06 | 87 [--] |
| - Observational evidence | 2 | 167/407 vs. 132/294 | 1.65 [0.23, 11.8] | 0.9 | 0 [--] |
| *By cervical length*: | | | | | |
| **Composite adverse outcome—<15 mm** | | | | | |
| - Observational evidence only | 2 | 54/104 vs. 85/114 | 0.75 [0.61, 0.92] | 0.03 | 4 [--] |
| **Composite adverse outcome—15–25 mm** | | | | | |
| - Observational evidence only | 1 | 14/41 vs. 22/30 | 0.47 [0.29, 0.75] | 0.002 | – |
| **RDS** | **3** | **29/128 vs. 15/122** | **2.32 [0.66, 8.10]** | **0.2** | **64 [0, 90]** |
| - Randomized evidence | 2 | 14/86 vs. 12/76 | 1.03 [0.51, 2.08] | 0.9 | 0 [--] |
| - Observational evidence | 1 | 15/42 vs. 3/46 | 4.78 [1.65, 13.8] | 0.004 | – |
| **Sepsis** | | | | | |
| - Randomized evidence only | 2 | 0/42 vs. 2/46 | 0.54 [0.07, 3.96] | 0.5 | 0 [--] |
| **Grades 3–4 IVH** | **3** | **4/128 vs. 5/122** | **0.85 [0.25, 2.88]** | **0.8** | **0 [0, 90]** |
| - Randomized evidence | 2 | 1/42 vs. 3/46 | 0.56 [0.10, 3.06] | 0.5 | 0 [--] |
| - Observational evidence | 1 | 3/86 vs. 2/76 | 1.33 [0.23, 7.69] | 0.8 | – |

*(Continued)*

**Table 7.** (Continued)

| Outcomes | Number of studies | Total women n/N vs. n/N | RR [95% CI] | p Value | I² [95% CI], % |
|---|---|---|---|---|---|
| **Birthweight <1,500 g—all studies** | **3** | **139/416 vs. 105/312** | **1.53 [0.51, 4.59]** | **0.5** | **81 [41, 94]** |
| - Randomized evidence | 2 | 19/42 vs. 7/46 | 2.73 [1.00, 7.42] | 0.05 | 23 [––] |
| - Observational evidence | 1 | 120/374 vs. 98/266 | 0.87 [0.70, 1.08] | 0.2 | – |
| **NICU admission—all studies** | **3** | **2036/423 vs. 170/312** | **0.90 [0.77, 1.07]** | **0.2** | **12 [0, 91]** |
| - Randomized evidence | 1 | 5/16 vs. 9/18 | 0.63 [0.26, 1.48] | 0.3 | – |
| - Observational evidence | 2 | 201/407 vs.161/294 | 0.92 [0.76, 1.12] | 0.4 | 37 [––] |

*N. of fetuses.

RR, risk ratio; CI, confidence interval; n/N vs. n/N, number of women with the outcome/total number of women in the exposed [cerclage] and unexposed [no cerclage] group, respectively; pPROM, preterm premature rupture of membranes; IVH, intraventricular hemorrhage; NICU, neonatal intensive care unit; RDS, respiratory distress syndrome.

In the current review, we have also confirmed the potential beneficial role of cerclage in reducing the risk of PTB and neonatal morbidity in twin pregnancies with a cervical length <15 mm, similar to that reported in singleton pregnancies. Conversely, in women with a cervical length of 15 to 25 mm, cerclage was not associated with a reduction in the risk of any of the outcomes assessed. These findings are consistent with those of studies on the predictive accuracy of ultrasound in twin pregnancies that report that lower cut-offs of cervical length compared to those used in singletons better predict PTB in multiple gestations [57]. Mid-trimester mean cervical length is less in twin compared to singleton gestations and it is biologically plausible that this reduction may be due to the effect of uterine overdistension on the cervix, leading to a relative shortening compared to singletons, without being associated with an increased risk of PTB. On this basis, placement of cervical cerclage in women with cervical length >15 mm on ultrasound should not be recommended, although these findings are based on observational evidence.

Although the findings from this meta-analysis suggest a potential beneficial role of cervical cerclage in reducing the risk of PTB and improving neonatal outcome in women at risk, the inclusion of mainly observational studies significantly affect the robustness of the results and should be confirmed in adequately powered RCTs. Only 3 RCTs were included, with a very small number of women allocated to cerclage or standard care. Ideally, an RCT of the role of cerclage in twin pregnancies should include women with short cervix on ultrasound or cervical dilatation separately and be adequately powered to investigated maternal and neonatal outcomes. Furthermore, this trial should share an objective protocol of prenatal assessment of women at risk and management of women before and after cerclage placement, including the timing of ultrasound assessment of cervical and preventive strategies of PTB, including progesterone and tocolysis.

Twin pregnancies undergoing cerclage for short cervix at ultrasound or cervical dilatation at physical examination have a lower risk of PTB and perinatal mortality and morbidity compared to those not undergoing such intervention. However, these findings are driven mainly from observational studies, thus limiting the robustness of the results. The findings from the present systematic review highlight the need for designing an appropriately powered RCT to elucidate whether the placement of a cervical cerclage may prevent PTB in women presenting with short cervical length at ultrasound assessment or cervical dilatation at physical examination.

## Supporting information

**S1 Table. Search strategy.**
(DOCX)

**S2 Table. Prisma checklist.**
(DOCX)

**S3 Table. Excluded studies and reason for exclusion.**
(DOCX)

**S4 Table. Pooled proportions for the perinatal outcomes explored in the present systematic review (95% confidence intervals between parentheses) in twin compared pregnancies undergoing compared to those not undergoing cerclage.**
(DOCX)

**S5 Table. GRADE assessment of the primary outcome.**
(DOCX)

**S1 Fig. Funnel plot of the effect estimates vs. their standard errors (outcome: risk of preterm birth <34th week in women undergoing cerclage versus no cerclage—women with a reduced cervical length on ultrasound and/or cervical dilatation at examination).**
(DOCX)

**S2 Fig. Funnel plot of the effect estimates vs. their standard errors (outcome: risk of preterm birth <32nd week in women undergoing cerclage versus no cerclage—women with a reduced cervical length on ultrasound and/or cervical dilatation at examination).**
(DOCX)

**S3 Fig. Funnel plot of the effect estimates vs. their standard errors (outcome: risk of preterm birth <28th week in women undergoing cerclage versus no cerclage—women with a reduced cervical length on ultrasound and/or cervical dilatation at examination).**
(DOCX)

**S4 Fig. Funnel plot of the effect estimates vs. their standard errors (outcome: gestational age in women undergoing cerclage versus no cerclage—women with a reduced cervical length on ultrasound and/or cervical dilatation at examination).**
(DOCX)

**S5 Fig. Funnel plot of the effect estimates vs. their standard errors (outcome: gestational age in women undergoing cerclage versus no cerclage—women with a reduced cervical length on ultrasound).**
(DOCX)

**S6 Fig. Funnel plot of the effect estimates vs. their standard errors (outcome: presentation to delivery interval in women undergoing cerclage versus no cerclage—women with a reduced cervical length on ultrasound and/or cervical dilatation at examination).**
(DOCX)

**S7 Fig. Funnel plot of the effect estimates vs. their standard errors (outcome: birthweight in women undergoing cerclage versus no cerclage—women with a reduced cervical length on ultrasound and/or cervical dilatation at examination).**
(DOCX)

## Author Contributions

**Conceptualization:** Francesco D'Antonio, Lamberto Manzoli.

**Data curation:** Francesco D'Antonio, Nashwa Eltaweel, Lamberto Manzoli, Asma Khalil.

**Formal analysis:** Asma Khalil.

**Investigation:** Francesco D'Antonio, Maria Elena Flacco.

**Methodology:** Francesco D'Antonio, Nashwa Eltaweel, Smriti Prasad, Maria Elena Flacco, Lamberto Manzoli, Asma Khalil.

**Project administration:** Lamberto Manzoli.

**Resources:** Lamberto Manzoli.

**Software:** Maria Elena Flacco, Lamberto Manzoli.

**Supervision:** Smriti Prasad, Maria Elena Flacco, Lamberto Manzoli.

**Validation:** Maria Elena Flacco, Lamberto Manzoli.

**Visualization:** Lamberto Manzoli, Asma Khalil.

**Writing – original draft:** Francesco D'Antonio, Nashwa Eltaweel, Smriti Prasad, Maria Elena Flacco, Lamberto Manzoli, Asma Khalil.

**Writing – review & editing:** Francesco D'Antonio, Nashwa Eltaweel, Smriti Prasad, Maria Elena Flacco, Lamberto Manzoli, Asma Khalil.

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
