## [Editor Report · Decision Letter 0]

29 Dec 2022

Dear Dr D'Antonio, 

Thank you for submitting your manuscript entitled "Cervical cerclage for prevention of preterm birth and adverse perinatal outcome in twin pregnancies: Systematic Review and Meta-analysis" for consideration by PLOS Medicine.

Your manuscript has now been evaluated by the PLOS Medicine editorial staff and I am writing to let you know that we would like to send your submission out for external peer review.

Please re-submit your manuscript within two working days, i.e. by Jan 02 2023 11:59PM.

Kind regards,

Philippa Dodd, MBBS MRCP PhD

PLOS Medicine

---

## [Decision Letter · Decision Letter 1]

21 Mar 2023

Dear Dr. D'Antonio,

Thank you very much for submitting your manuscript "Cervical cerclage for prevention of preterm birth and adverse perinatal outcome in twin pregnancies: Systematic Review and Meta-analysis" (PMEDICINE-D-22-03921R1) for consideration at PLOS Medicine. 

[LINK]

In light of these reviews, I am afraid that we will not be able to accept the manuscript for publication in the journal in its current form, but we would like to consider a revised version that addresses the reviewers' and editors' comments. Obviously we cannot make any decision about publication until we have seen the revised manuscript and your response, and we plan to seek re-review by one or more of the reviewers. 

We expect to receive your revised manuscript by Apr 11 2023 11:59PM. Please email us (plosmedicine@plos.org) if you have any questions or concerns.

We look forward to receiving your revised manuscript. 

Sincerely,

Philippa Dodd, MBBS MRCP PhD

PLOS Medicine

plosmedicine.org

GENERAL

Please respond to all editor and reviewer comments detailed below, in full.

Thank you for reporting your SRMA according to PRISMA. Please provide the completed PRISMA checklist. 

When completing the checklist, please use section and paragraph numbers, rather than page or line numbers as these often change in the event of publication.

Please modify the statement in the Methods section to signpost the checklist: "This study is reported as per the Preferred Reporting Items for Systematic Reviews and Meta-Analyses (PRISMA) guideline (S1 Checklist)." Or similar.

*** The reviewers and academic editor have raised specific concerns regarding the design and methodological approaches used in your study. Please see below for specific comments which we agree with. ***

COMMENTS FROM THE ACADEMIC EDITOR

1. The paper cannot be published in its present form.

2. The authors seem to interpret their results as making the case for cerclage in all twins with a short cervix (<15mm). But the vast bulk of the data come from observational studies and, as the stats review points out, the RCTs differ from the observational studies and there is obviously huge potential for bias when comparing women having and not having an intervention in an observational study design.

3. Even the RCT data are poor. Much of it is old, and even the more recent Roman paper only included 34 women and the intervention also included indomethacin and antibiotics, so it is not clear that any effect in this group was due to cerclage.

I think that this paper best serves to provide the data required to design the RCT that could change clinical practice. I recommend:

1. That they need to remove all analyses where they just lump all twin pregnancies together, i.e. there should be separate analysis of those where cerclage was indicated and those where it was not and no analysis where these two groups are pooled.

2. That they provide separate analysis of RCTs and observational studies and they should formally test for differences in the estimate of effect size between the two groups.

3. They re-focus the discussion around how their analyses inform a trial - inclusion criteria, comparison group, primary outcome, sample size etc.

ABSTRACT

Please ensure that your abstract is reported according to PRISMA for abstracts, following the PLOS Medicine abstract structure (Background, Methods and Findings, Conclusions) http://www.plosmedicine.org/article/info:doi/10.1371/journal.pmed.1001419

Abstract methods and findings: this reads very nicely but does seem to be rather long. Perhaps fewer of the secondary outcomes could be listed (those considered most important) and the same examples then evidenced with statistical information.

Please provide the beginning and dates of the (updated) search, data sources, number of studies included, types of study designs included, and synthesis/appraisal methods 

Line 85: “The level of evidence was downgraded…” the sentence is rather vague (“imprecision of results” and “indirectness of evidence”) and somewhat discredits your own data. Suggest revising this statement and concluding this section with a sentence that details the main limitations of your study.

AUTHOR SUMMARY

METHODS and RESULTS

Please update your search to the present time. We require that SRMAs are updated to within roughly 6 months of the expected publication date.

Please provide the beginning and end dates of your search.

Please also note the comments from the methodological reviewer (reviewer #1) regarding inclusion of non-English language sources of studies which we agree with. Please include non-English language sources of studies in your search.

Thank you for reporting your SRMA according to PRISMA. Please remove the repeated statement at lines 132-133.

TABLES

To help facilitate transparent data reporting, PLOS medicine requests that where adjusted analyses are presented unadjusted analyses are also presented for comparison. We understand from your methods section that you extracted a combination of adjusted and unadjusted estimates from the studies included in your meta-analysis. We appreciate that this may not be feasible (or necessary) to present both in this case but it may be helpful if you could indicate which of the extracted data are adjusted and which are not and if the data are available might it also be helpful to include factors that were adjusted for? Perhaps in table 1?

When reporting p values please report as p<0.001 or where higher as p=0.002, for example. Please check and amend throughout including in the supporting files where relevant. 

SUPPORTING INFORMATION

Table S4 – is there a typo in column 3 (chorioamnionitis RCTs)?

DISCUSSION

Please remove all sub-headings from the discussion such that it reads as a single piece of continuous prose starting with a short, clear summary of the article's findings; what the study adds to existing research and where and why the results may differ from previous research; strengths and limitations of the study; implications and next steps for research, clinical practice, and/or public policy; and ending in a one paragraph conclusion.

Comments from the reviewers:

Reviewer #1: See attachments

Michael Dewey

Reviewer #2: The paper presents a systemic review and meta-analysis of cervical cerclage for prevention of preterm birth and adverse perinatal outcome in twin pregnancies. The description of the study rationale, the interpretation of the results and the discussion are balanced. However, similar articles have been published in the peer-reviewed scientific literatures such as,

Su J, Li D, Yang Y, Cao Y, Yin Z. Cerclage placement in twin pregnancies with cervical dilation: a systematic review and meta-analysis. J Matern Fetal Neonatal Med. 2022 Dec;35(25):9112-9118. doi: 10.1080/14767058.2021.2015577. Epub 2021 Dec 14. PMID: 34906023.

Li C, Shen J, Hua K. Cerclage for women with twin pregnancies: a systematic review and metaanalysis. Am J Obstet Gynecol. 2019 Jun;220(6):543-557.e1. doi: 10.1016/j.ajog.2018.11.1105. Epub 2018 Dec 7. PMID: 30527942.

Li C, Hua K. Efficacy of physical examination-indicated cerclage in twin pregnancies compared with singleton pregnancies: a systematic review and meta-analysis. Minerva Obstet Gynecol. 2021 Feb;73(1):111-120. doi: 10.23736/S2724-606X.20.04518-9. Epub 2020 Apr 21. PMID: 32315128.

Saccone G, Rust O, Althuisius S, Roman A, Berghella V. Cerclage for short cervix in twin pregnancies: systematic review and meta-analysis of randomized trials using individual patient-level data. Acta Obstet Gynecol Scand. 2015 Apr;94(4):352-8. doi: 10.1111/aogs.12600. Epub 2015 Mar 1. PMID: 25644964.

The authors should mention the previous studies and discuss the additional information of this manuscript compared with previous studies. Besides, PPROM should be preterm prelabor rupture of membranes (ACOG Practice Bulletin No. 217: Prelabor Rupture of Membranes. Siegler Y, et al. Obstet Gynecol. 2020. PMID: 33093409).

Reviewer #3: This systematic review addresses the important question of cerclage in twins. Historically cerclage has been said to be contraindicated, and this was based on a small meta analysis by Berghella that was under powered to inform on the rarer women with a very short cervical length. Indeed, in the UK, NICE, as an arbitor of clinical practice, until recently specifically advised against cervical scanning in women with twins. More recently a number of small series or trials in various heterogeneous groups have presented data that challenges this.

A synthesis of the results looks at

1. Overall, 'in pregnancy' cerclage (<25mm or physical exam), or whatever the authors wish to call it, but not emergency cerclage. 

Subgroups of indication for 'in pregnancy' cerclage: 

2. Cervical length <15 mm 

3. Physical examination 

Then:

4. Pessary (without conclusions)

5. Twins versus singletons

This is a highly appropriate paper and should be published. 

Points.

1. I am not an expert in the methodology and so am not qualified to comment on this.

2. The authors have confused me on definitions. They define emergency cerclage as that where the cervix is <25mm or 'open' physical examination. This is not correct by conventional terminology. It is not unreasonable to group these together but this perhaps should be defined as non-elective or 'short cervix/ physical examination'. 

3. Even though putting cervical cerclage at 0mm cervical length, short cervical length +/- with bulging membranes together with 'physical examination indicated' is pragmatic, it is still a bit problematic because of the heterogeneity: the US RCT of physical examination indicated cerclage (Roman et al) did not specify cervical length in all women and some were clearly not that short. The heterogeneity of this group should be emphasised, and perhaps a later comment about the different practice in Europe (cervical length) and the US (physical examination)

4. Line 302: 'Conversely, cerclage in women with a cervical length between 5 and 25 mm was not.. ' This is a typo? Should 5 not be 15?

5. I would like to see data in the text as to effect sizes particularly whether any outcomes were increased (as opposed to not reduced) in the 15-25mm cervical length group. 

6. Could the authors differentiate between monochorionic and dichorionic twins?

7. Given the importance of the findings and the need for evidence-based practice I would like to see, in the conclusions, a recommendation regarding best practice, including recommending against cervical cerclage where the cervix is >25mm. 

Reviewer #4: The authors present a meta-analysis of both observational and randomised control trials of cerclage in twins. This is in a currently challenging area with considerable uncertainty and variation in clinical practice. The topic is important and worthy of publication. However we believe that more data could be derived from their analysis that would have important clinical implications.

The authors have subdivided their groups for analysis into two main categories:

Those with dilated cervix and/or those with a shortened cervix <25mm

Elective cerclage (who have presumably not got a short cervix, but this is not clear)

Those in the first group may be substantially different, depending on whether the cervix id dilated or not, and cerclage could have varying treatment affects in these groups. A dilated cervix with exposed membranes will allow immediate ascending infection and a cerclage in effect is trying to reverse a high risk situation. However the group with a cervix <25mm may be upstream in the pathophysiological process and a cerclage is more preventative. It is unclear why they have combined these groups. Indeed their sub-analysis demonstrates a difference in those with cervix >15mm and <15mm, suggesting the concerns above are valid. We are a little concerned that this combination may have been performed to avoid a type II error in one of these sub-groups.

We strongly recommend primary analysis is performed separately for women with dilated cervix and in those with a short cervix, as well as those with an elective procedure (i.e. three groups in total). Or alternatively reasons for not doing this described.

An additional analysis which would be clinically very relevant would be those with symptomatic and asymptomatic presentations. Outcomes following emergency cerclage are very different if women present with discharge and pain at presentation and subsequently had a cerclage versus those who were incidentally found to have a short cervix. We don't feel as strongly about this point but some exploratory analysis whether this clinical factors are important, should they be available in their data, would be worthwhile.

The terminology used for elective cerclage is 'unselected population'. This is not a commonly used terminology and is ambiguous as it implied all comers including those with a short cervix, etc. Again this should be clarified and we believe the term is elective cerclage is more ubiquitous.

In the group with elective cerclage, a problem with all research in twin pregnancies is that low risk women are often recruited, as they are easy to obtain for clinical trials. However, are there are sound pathophysiological reasons why women with risk factors may behave differently. At very least they should discuss if studies evaluated these sub-groups and ideally perform a separate analysis if the data if the data is available. 

A general comment is that it may be wise to improve the readability of the article, as we must confess to having to read several paragraphs multiple times. We appreciate the challenge of writing in a language that is not one's native language, however the team will likely have the expertise to improve this. 

This is an important piece of work and the team have done well with good analysis and correct endpoints. We believe with extra analysis, significantly more clinically relevant findings could be obtained (if the data is available). We strongly recommend this with a view to publication. 

Reviewer #5: nicely written concise systematic review with sound methodology and appropriately utilised tools for the meta-analysis. 

Reviewer #6: This paper is confusing. 

My main concern is that the authors pool everything together and this does not facilitate using this paper in clinical practice (see also comments below).

They pool cases with an ultrasound indicated cerclage with those of physical examination indicated cerclage in patients with cervical dilatation, generally referred to as 'emergency cerclage'. I advise you to differentiate between those 2 groups.

Moreover, as stated in sentence 136 and 137: inclusion criteria were studies in which twin pregnancies were allocated to cerclage for the prevention of PTB or to a control group (e.g. placebo or treatment as usual). Thus the studies comparing with pessary should not be included, this is confusing. 

The paper would benefit if the authors would put their results into more context (and not just a repetition that preterm birth is a large problem....). But what does this study add. How do we proceed from here? What is needed? I miss some depth in the discussion. 

Abstract

-The abstract is a bit long; consider reporting only the primary outcome and the most important secondary outcomes.

Introduction

- Sentence 95: 'The risk' probably refers to the risk of perinatal morbidity and mortality, this is however not very clear. In addition, I would recommend adding a reference to this sentence. 

- Sentence 103: If this sentence refers to singleton pregnancies, I would advise you to state this more clearly and choose a different study to refer to as this study is meanly about twin pregnancies. 

Methods

- Sentence 136: Add to the inclusion criteria that you included only randomized controlled trials (RCTs), prospective studies, and retrospective studies.

- Describe your secondary outcomes in a sentence rather than point by point.

- Generally 'Emergency cerclage' is defined as cerclage placed for cervical dilatation at a physical examination. An ultrasound indicated cerclage is placed for a short cervix (<25mm) at the ultrasound. To prevent confusion, please consider describing these groups separately as 'ultrasound indicated cerclage' and 'physical examination indicated cerclage'. 

- Which software was just for the statistical analyses? 

- As stated in the article, adjusted data was extracted from the studies or, when these were not available, the unadjusted estimates. (is this correct: question for statistician).

Results

- Check if Figure 1 is correct, in the text it is stated that 38 studies were included in the meta-analyses while the figure shows 53 studies. 

- If the inclusion criteria for the review were studies in which women with a cerclage were compared to a control (placebo or treatment as usual), why were two studies comparing a cerclage to a pessary included? I would exclude these studies from your review because this is not relevant to this article. 

- I would advise you to switch paragraphs 2 (from sentence 244) and three (from sentence 253) because paragraph three (quality assessments) is still part of your study selection. 

- Please check all the table numbers as you have two tables 2. 

- Sentence 256: Is the ROBINS-I tool used (as described here) to access the risk of bias in the observational studies or, as described in the methods, the NOS?

- Sentence 271. The p-value in the text does not match the one in the table. (table 0,011 vs. text 0,001)

- It is expected that women with a short cervix and cervical dilatation have a different risk of PTB. I therefore recommend you to analyze / describe the results of these groups only separately. Remove table 3. Describe the results of women with a short cervix (now table 5) and women with cervical dilatation (now table 6) more comprehensive. Remove table 4 and add this information to the other tables. 

- You made two forest plots (PTB <34 weeks and composite morbidity), however, this is not mentioned in the text. 

- Was a sensitivity analysis done?

Discussion

- The paragraph 'interpretation of results ….' is missing the research implications: How can future research build on these observations and what are the key experiments that must be done?

- I miss the interpretation of the results paragraph. For example, you report that no difference in PPROM or Chorioamnionitis was found between women who received an emergency cerclage and women with expected management. Can you explain this? Was PPROM ruled out before cerclage placement in some of the studies? Did all studies describe the use of antibiotics or tocolytics?

- How do the conclusions of your review affect the existing knowledge in the field?

- Sentence 381: add a reference.

- Sentence 430 is hard to read.

[LINK]

---

## [Decision Letter · Decision Letter 2]

6 Jun 2023

Dear Dr. D'Antonio,

Thank you very much for re-submitting your manuscript "Cervical cerclage for prevention of preterm birth and adverse perinatal outcome in twin pregnancies with short cervical length or cervical dilatation: a systematic Review and Meta-analysis" (PMEDICINE-D-22-03921R2) for review by PLOS Medicine.

I have discussed the paper with my colleagues and the academic editor and it was also seen again by 3 reviewers. I am pleased to say that provided the remaining editorial and production issues are dealt with we are planning to accept the paper for publication in the journal.

[LINK]

We look forward to receiving the revised manuscript by Jun 13 2023 11:59PM.   

Sincerely,

Philippa Dodd, MBBS MRCP PhD

Senior Editor 

PLOS Medicine

plosmedicine.org

Requests from Editors:

GENERAL

Thank you for your detailed and considered responses to previous editor and reviewer comments.

Please see below for further comments which we require you address in full, prior to publication.

Throughout, please replace the term ‘retrospective’ with ‘observational’ when describing your study. 

STATISTICAL REPORTING

Suggest separating upper and lower bounds of CIs with commas as opposed to hyphens which can be confused with reporting of negative values. 

PLOS requests that where 95% CIs are reported p values are also reported, please include. Please report p values as <0.001 and where higher as p=0.002, for example. 

Suggest reporting statistical information as follows, ‘(RR: 0.73, 95% CI [0.59,0.91], p=/<)’ for improved reader accessibility. Please check and amend throughout the main manuscript, tables and figures where relevant, including in the supporting information.

ABSTRACT

Line 47 – please expand the background to (briefly) provide wider context of why the study question is important. The statement currently written at line 47 should constitute the last sentence of the Abstract Background.

Line 51 – thank you for updating your search. Please include the search start date – e.g. inception or a specific date and in the event that you report a specific date please briefly justify the reasons for its choice.

Line 65 onward - please include the actual amounts and/or absolute risk(s) of relevant outcomes, not just relative risks or correlation coefficients (example for absolute risks: PMID: 28399126).

Line 72 onward – as above, suggest separating upper and lower bounds of CIs with commas as opposed to hyphens which can be confused with reporting of negative values. PLOS requests that where 95% CIs are reported p values are also reported, please include. Please report p values as <0.001 and where higher as p=0.002, for example. 

Line 74 – please define MD (? Mean difference) prior to first use here – perhaps at line 66?

AUTHOR SUMMARY

Thank you for including an author summary. Please see below for suggested revisions to formatting and minor changes to the content:

Why Was This Study Done? 

* Twin pregnancies are at higher risk of preterm birth (PTB). 

* Recent evidence suggests that placement of cervical cerclage in twin pregnancies with short cervical length at ultrasound or cervical dilatation at physical examination might be associated with a reduced risk of PTB. 

* However, such evidence is based mainly on small studies thus questioning the robustness of these findings. 

What Did the Researchers Do and Find? 

* We performed a systematic review and meta-analysis to elucidate whether cervical cerclage in women with twin pregnancy with short cervical length or cervical dilatation may prevent PTB. We included eighteen studies. The primary outcome was PTB <34 weeks of gestation. 

* We found that cervical cerclage in women with short cervical length or cervical dilatation was associated with a reduced risk of PTB <34 weeks, gestational age at birth and adverse neonatal outcome.

* The strength of association between cerclage and reduced risk of PTB was maintained when considering women with short cervix on ultrasound and those with cervical dilatation at physical examination separately. 

What Do These Findings Mean? 

* Cervical cerclage in twin pregnancies with short cervical length or cervical dilatation may be potentially associated with a reduced risk of PTB and improved neonatal outcomes. 

* However, these findings are mainly based on observational studies and, to improve robustness of evidence, confirmation of these outcomes in large and appropriately designed RCTs is required.

METHODS and RESULTS

As for the abstract please include details of the beginning dates of your search.

Line 192 – ‘(McDonald vs Shirodkar)’ please indicate for the reader that these are different techniques and the most utilised approaches for cerclage.

Line 252-3 – please apply PLOS Medicine’s required referencing format here.

Line 267 onward – please use either numbers or words to depict numerical values as opposed to a combination of both.

Lines 288-290 – see above under statistical reporting. Here you use the word ‘to’ to separate upper and lower bounds in view of negative values, suggest instead using commas throughout.

Line 293 onward – please amend statistical reporting as detailed above. Please check and amend throughout.

Line 308 – ‘pPROM’ please ensure this has been defined at first use for the reader – apologies if I have missed it.

TABLES

Table 1 – column ‘Gestational age at cerclage placement’ please define the age measurements (weeks/days). Please define the numerical values contained within parentheses. Does the +/- refer to days or weeks or both? Throughout, please define ‘NR’ for the reader. Please replace ‘retrosp.’ with ‘OBS’ or similar to describe the studies as observational and please define the abbreviation in the caption for the reader. 

Table 3 – please indicate if more (or less) stars equate to higher (or lower) quality.

Table 4 – throughout where reporting 95% CIs please separate upper and lower bounds with commas (not semi-colons or hyphens). Is there room to write ‘Number of studies’ in the column 2 header? Please change the penultimate column header to ‘p value’. 

Table 5, 6, 7 & 8 – as above

DISCUSSION

Line 466 – please remove the sub-heading ‘Conclusions’

Line 474 – please remove the funding statement and include only in the manuscript submission form when you resubmit the manuscript, it will be compiled as metadata at the time of publication.

REFERENCES

Ref 28 – appears incomplete

SUPPORTING INFORMATION

Thank you for including the PRISMA checklist. Please revise to refer to section and paragraph numbers as opposed to page (and/or line) numbers as these often change at publication.

SOCIAL MEDIA

If not already done so, to help us extend the reach of your research, please detail any Twitter handles you wish to be included when we tweet this paper (including your own, your coauthors’, your institution, funder, or lab) in the manuscript submission form when you re-submit the manuscript.

Comments from Reviewers:

Reviewer #1: The authors have addressed all my points.

Michael Dewey

Reviewer #3: The heterogeneity of studies included (observational and randomised; physical exam indicated, emergency and cervical length/ different cervical lengths) were always going to make this analysis difficult and limit its conclusions.

The authors have performed considerable work to address the issues raised by the multiple reviewers and I believe have done as good a job as is possible. Although I remain slightly concerned about the different conclusions regarding observation and randomised studies with the statistical reviewer 1, I am not convinced this matters too much as the conclusions are clear that the 'positive' findings apply to observational rather than randomised studies. This limits their robustness but again the authors are clear about this.

Reviewer #6: Nicely performed systematic review. I am satified with the modifications made.

[LINK]

---

## [Editor Report · Decision Letter 3]

23 Jun 2023

Dear Dr D'Antonio, 

On behalf of my colleagues and the Academic Editor, Professor Gordon Smith, I am pleased to inform you that we have agreed to publish your manuscript "Cervical cerclage for prevention of preterm birth and adverse perinatal outcome in twin pregnancies with short cervical length or cervical dilatation: a systematic Review and Meta-analysis" (PMEDICINE-D-22-03921R3) in PLOS Medicine.

Prior to publication we require that you address the following issues:

1) Please ensure the authors summary is divided into bullet points as requested previously. This is a formatting requirement

2) Line 361 – please replace hyphen with comma ‘95% CI [0.52-0.88]’ here

3) Table 4&5 – please replace uppercase P with lowercase p in column header

4) Bibliography - please ensure no more than 6 authors are listed followed by et al (ref 31 for example) 

5) PRISMA Checklist – please update the checklist to refer to section and paragraph numbers rather than page and/or line numbers as these often change at publication

6) Supplementary Table 1 – you updated your search as part of major revision but the table doesn’t reflect that please amend

7) Supplementary Table 4 – contains a header referring to supplementary table 3 please clarify/revise. Please also replace hyphens with commas when reporting CIs here

8) Supplementary figure 1 – please report lowercase p, please replace ‘to’ with hyphens when reporting CIs here

PRESS

Best wishes,

Pippa 

Philippa Dodd, MBBS MRCP PhD 

PLOS Medicine